# Coupled dynamics of resource competition and constrained entrances in a multi-lane bidirectional exclusion process

**Ashish Kumar Pandey**$^\star$ **and Arvind Kumar Gupta**$^\dagger$

Department of Mathematics, Indian Institute of Technology Ropar,
Rupnagar- 140001, Punjab, India

$\star$ ashish.23maz0003@iitrpr.ac.in,   $\dagger$ akgupta@iitrpr.ac.in

## Abstract

Motivated by the role of limited particle resources in multi-species bidirectional transport processes observed in various biological and physical systems, we investigate a one-dimensional closed system consisting of two parallel lanes with narrow entrances, where each lane accommodates two oppositely directed particle species. Each particle species is linked to a separate finite reservoir, which is coupled to both lanes and regulates the entry rates of particle into the lanes. To analyze the effect of finite particle reservoirs on the stationary properties of the system, we employ a mean-field theoretical framework to characterize the density profiles, particle currents, and phase behavior, complemented by a boundary layer analysis based on fixed point methods to capture spatial variations near the boundaries. The impact of individual species population, quantified by species-specific filling factors, is systematically examined under both equal and unequal conditions. For equal filling factors, system undergoes spontaneous symmetry breaking and supports both symmetric and asymmetric phases. In contrast, for unequal filling factors, only asymmetric phases are realized, with the phase diagram exhibiting up to five distinct phases. A striking feature observed in both scenarios is the emergence of a back-and-forth transition, along with a non-monotonic dependence of the number of phases on the filling factors. All theoretical findings are extensively validated through stochastic simulations based on the Gillespie algorithm, confirming the robustness of the analytical results.

# 1   Introduction

Stochastic transport phenomena in complex systems has emerged as a central subject of investigation in nonequilibrium statistical physics, owing to its fundamental importance and applicability across a wide range of physical and biological systems. Several transport processes including intracellular cargo transport along microtubules [1,2], vehicular traffic flow [3,4], and mRNA translation [5,6] have been effectively modeled by systems of driven particles undergoing stochastic motion along one-dimensional lattices. To analyze the driven dynamics of such systems, Totally Asymmetric Simple Exclusion Process (TASEP) has emerged as a paradigmatic model, widely employed to investigate a variety of stationary properties [7–9]. This model has been substantially utilized to examine the transport phenomena across a broad spectrum of systems, ranging from physical contexts such as pedestrian movement [10], vehicular traffic [3], and ant trail formation [11], to biological processes including the directed motion of motor proteins [12] and protein synthesis during mRNA translation [13]. Originally proposed in 1968 to model biopolymerization by ribosomes [5,6], TASEP has since evolved into a foundational tool for investigating nonequilibrium behavior in systems of interacting particles moving along one-dimensional lattices. In such models, the lattice is governed by distinct boundary conditions: generally open or periodic, which play a crucial role in shaping the system's dynamics and steady-state behavior. In the case of open boundaries, this model captures the collective dynamics of particles that enter and exit at the boundaries of a lattice and hop unidirectionally in the bulk with unit rate to the adjacent site, subject to the hard-core exclusion principle that prohibits multiple occupancy. In addition to exact methods [14], the mean-field approximation offers quantitatively accurate predictions for steady-state properties including bulk densities and currents that closely align with exact solutions. In the thermodynamic limit, the exact steady-state bulk currents are characterized by a phase diagram comprising three distinct regimes: low density (LD), high density (HD), and maximal current (MC) phases. Moreover, the boundary between the low density (LD) and high density (HD) phases corresponds to a coexistence line characterized by a delocalized shock, which

separates the regions of differing particle densities within the system [9]. To capture the full spatial structure of the density profile including the behavior near the boundaries where the bulk solution typically fails to satisfy all the imposed boundary conditions, analytical techniques such as singular perturbation theory along with fixed point analysis are employed to obtain boundary layers and characterize their structure [15]. Investigations of this straightforward model and its variants have uncovered a rich spectrum of nonequilibrium phenomena, including phase transitions induced by boundary and bulk effect, shock formation, phase coexistence and spontaneous symmetry breaking (SSB) [5,6,8,14,16–19].

In biological systems, microtubules and actin filaments play a fundamental role in intracellular transport, particularly in processes such as axonal transport within neurons [20]. These filaments form an extensive network that serve as tracks for motor proteins like kinesin and dynein, which move in opposite directions and are responsible for carrying various cellular cargos such as mitochondria, endosomes, lipid droplets, and even viral components to specific locations within the cell [21]. The efficient functioning of this motor protein driven system is vital for maintaining cellular function, as disruptions or genetic defects in these pathways have been associated with several diseases, including neurodegenerative disorders, Alzheimer's disease, hearing impairment, and polycystic kidney disease [22]. These motor proteins rely on the hydrolysis of ATP (adenosine triphosphate) to convert chemical energy into mechanical force, enabling active and directed movement along the cytoskeletal filaments toward specific ends. This directed motion of distinct motor proteins toward opposite ends of cytoskeletal filaments gives rise to multispecies bidirectional transport. Specifically, kinesin-1 travel toward the plus end of microtubules, typically directed toward the cell periphery, whereas dynein-1 move in the opposite direction, toward the minus end, which is generally located near the nucleus or cell center [20,21]. Such bidirectional transport is not only confined to biological systems but has also been manifested in a variety of engineered systems, including vehicular traffic and pedestrian movement [3,23–27].

Numerous studies have been devoted to generalizing the TASEP framework from single-species to multi-species systems, wherein two distinct particle types move in opposite directions on the same lattice [28,29]. Unlike the single-species case, these extensions exhibit a range of cooperative phenomena, including spontaneous symmetry breaking, phase separation and back-and-forth transitions [19,28–31]. One of the earliest models to exhibit symmetry broken phases in a bidirectional setting was the "bridge model" which featured oppositely directed particles on a single lane [32]. While this model marked a significant advance in the understanding of symmetry breaking in nonequilibrium systems, the exact mechanisms underlying the emergence of these broken-symmetry phases remain incompletely understood, and continues to be actively studied [32–34]. Subsequent works extended the bridge model to more complex geometries by introducing junction based structures [35], and further generalized it to incorporate the combined effects of stochastic directional switching and inter-lane transitions in multispecies, conserved two-channel TASEP systems [36].

Microtubules, beyond serving as transport pathways for motor proteins, possess structural features that significantly influence intracellular transport dynamics. Their protofilament architecture exhibits intrinsic helical twisting and curvature, which can lead to non-linear geometries such as circular or wavy trajectories [37]. These curved conformations naturally give rise to spatial constraints near filament junctions and terminal regions, effectively creating narrow entrances that function as bottlenecks within the transport network [38]. Such localized constraints and traffic congestion points can obstruct unidirectional flow and hinder bidirectional movement, making them key factors in regulating transport efficiency. These observations have inspired the development of theoretical mod-

els involving bidirectional transport on two parallel lanes, where particles move in opposite directions and interact only at the boundaries [39, 40]. Subsequent extensions have demonstrated similar phenomena in models with periodic and multilane geometries [41], as well as in cylindrical lattice structures with entrance constraints [42].

Most of the open-boundary TASEP studies involving multispecies systems have assumed coupling to reservoirs with infinite particle supply, an idealization that deviates significantly from conditions observed in real world systems. In contrast, a wide range of biological and physical transport processes such as ribosome translation, motor driven intracellular transport, pedestrian movement, and vehicular traffic operate under resource limited conditions, whether on single lane or multi-lane geometries [39, 43–48]. To capture such realistic constraints, various extensions of the TASEP model have introduced finite reservoirs, wherein the particle injection rate dynamically depends on the reservoir's occupancy. This mechanism leads to several nontrivial effects, including the formation of localized shocks in the density profile and strong retaliation between the reservoir and the system dynamics [26, 29, 39, 43–56]. Most of studies in this context have primarily focused on models where global conservation is imposed on the total particle number across all species, effectively treating the reservoir as a shared pool. However, this assumption can yield physically inconsistent results for instance, if the reservoir contains no particles of a particular species, yet the entry rate remains nonzero due to dependence on the total particle count allowing particles of a particular species to enter the system even when these are not present in the reservoir. Such scenarios violate species-level conservation and imply unphysical particle insertion. A more consistent alternative is to impose species-wise conservation, where the total number of particles for each species is independently constrained. This refinement not only eliminates the aforementioned inconsistencies but also reveals a variety of rich and nontrivial collective behaviors, revealing new aspects of the system's nonequilibrium dynamics [29, 55].

Motivated by the emergence of complex phenomena in multispecies transport systems, the present study investigates a more realistic two-lane exclusion process. The model features bidirectional movement of two species along parallel lanes with narrow entrances, where particle injection is regulated by the occupancy of corresponding boundary site on the adjacent lane, introducing inter-lane interactions localized at the boundaries. Additionally, each lane is strategically coupled to separate finite capacity reservoirs for each particle species. Our primary objective is to explore how the interplay between finite resources and narrow entrance constraints influences the stationary properties of the system. In particular, we investigate the emergence of collective phenomena, such as spontaneous symmetry breaking and phase separation. We construct a theoretical framework based on the mean-field approximation to analyze the bulk dynamics of bidirectional system and further employ fixed point analysis to investigate the formation and position of boundary layers, providing a detailed understanding of the steady-state density profiles and the phase behavior of the system.

The structure of the article is organized as follows: Section 2 provides a extensive description of the model, including the dynamic rules that govern the system. Furthermore, the theoretical framework of the mean-field approximation is delineated, which is utilized in Sec. 3 to obtain the existence condition for phases. Section 4 is dedicated to a thorough analysis of the system's stationary behavior, including phase diagrams, spontaneous symmetry breaking, phase coexistence, and a detailed examination of boundary layer formation via fixed point analysis. Finally, Sec. 5 provides a comprehensive summary of the proposed model along with key outcomes and potential directions for future research.

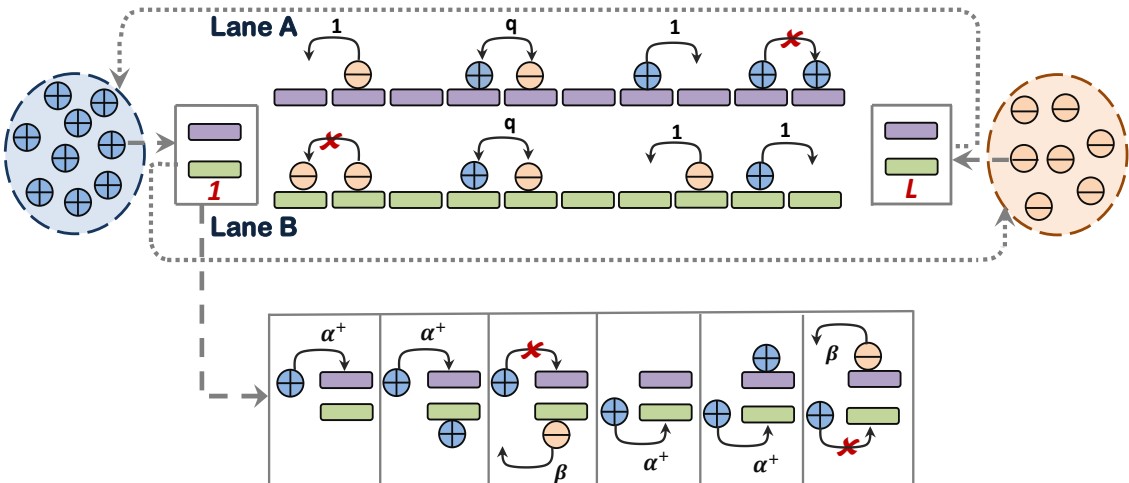

Figure 1: Schematic diagram of a two-lane bidirectional transport model, where each lane is connected to two finite reservoirs, accommodating particles of a single species only. Blue and orange circles represent oppositely directed $+$ and $-$ particles, entering with rates $\alpha^+$ and $\alpha^-$, and exiting with rate $\beta$. Particles of different species may exchange positions with rate $q$ when adjacent. Boundary dynamics at site $L$ is same as that on site 1, with reversed particle types.

## 2 Model description

We consider a system composed of two parallel one-dimensional lanes, denoted as lane A and lane B, each comprising $L$ discrete lattice sites labeled sequentially by indices $i = 1, 2, \ldots, L$. The sites $i = 1$ and $i = L$ represent the boundaries of each lane, while the intermediate sites with $1 < i < L$ constitute the bulk of the system. The system consists of two different species of particles, denoted as positive $(+)$ and negative $(-)$, which are distributed across both the lanes. These particles obey the hard-core exclusion principle, which ensure that each site on either lane can be occupied by at most one particle, regardless of its species, at any given time. As illustrated in Fig. 1, $+$ particles traverse from left to right along both lanes, while $-$ particles propagate in the opposite direction. Each particle species advances at unit rate in its respective direction, provided the adjacent site is vacant, thereby establishing bidirectional transport within each lane. When particles of distinct species encounter each other on the same lane, they exchange their positions with a rate $q$, provided the exchange is compatible with their respective directions of motion.

To reflect real-world scenarios involving limited resources, the boundaries of the lanes are coupled to two finite capacity reservoirs, $R^+$ and $R^-$, which possess no internal dynamics. The reservoir $R^+$ exclusively accommodates only $+$ particles, while $R^-$ solely sustains $-$ particles. Moreover both the reservoirs are assumed to be sufficiently large to accommodate all particles of their respective species present within the system. In particular, let $N^{\mathrm{tot}^+}$ and $N^{\mathrm{tot}^-}$ denote the total number of $+$ and $-$ particles in the system, respectively. Since the system is closed and both lanes are coupled to finite reservoirs, the total number of particles of each species in the system remains conserved at all times. This conservation condition can be expressed mathematically as:

$$N^{\mathrm{tot}^+} = N^{r^+}(t) + N^{A^+}(t) + N^{B^+}(t), \quad N^{\mathrm{tot}^-} = N^{r^-}(t) + N^{A^-}(t) + N^{B^-}(t). \quad (1)$$

Here, $N^{k^+}(t)$ and $N^{k^-}(t)$, where $k \in \{A, B\}$, denote the number of $+$ and $-$ particles in

lane $k$, while $N^{r^+}(t)$ and $N^{r^-}(t)$ denote the instantaneous number of $+$ and $-$ particles in the reservoirs $R^+$ and $R^-$, respectively at any given time $t$. This constraint on the particle count impacts the rates at which they enter the system from the reservoirs. As a result, the entry rates of $+$ and $-$ particles are not constant, they dynamically depend on the particle content of their respective reservoirs. It is clear that slower entrance rates correspond to a lower particle count in the reservoirs, whereas higher rates indicate a larger number of particles in the reservoirs. Therefore, the entrance rates [49] for the $+$ and $-$ species of particles are defined as follows:

$$\alpha^+ = \alpha\, f\big(N^{r^+}(t)\big), \quad \alpha^- = \alpha\, g\big(N^{r^-}(t)\big). \tag{2}$$

Here, $\alpha$ denotes the intrinsic entrance rate of particles, corresponding to the case of infinite particles reservoirs. The functions $f(\cdot)$ and $g(\cdot)$ are continuous, monotonically increasing functions that regulate the entry of $+$ and $-$ particles, respectively. As the simplest choice, these functions are defined as:

$$f\big(N^{r^+}(t)\big) = \frac{N^{r^+}(t)}{N^{\mathrm{tot}^+}}, \quad g\big(N^{r^-}(t)\big) = \frac{N^{r^-}(t)}{N^{\mathrm{tot}^-}}. \tag{3}$$

Clearly, $N^{r^+}(t) < N^{\mathrm{tot}^+}$ and $N^{r^-}(t) < N^{\mathrm{tot}^-}$, which implies that modified entrance rates are confined between 0 and $\alpha$.

Drawing motivation from collective transport along curved microtubules and analogous congestion phenomena in traffic flow, we consider a two-lane system with narrow entrances, where interactions between lanes occur exclusively at the boundaries. In particular, the entry of a particle into a lane depends on the occupancy of the corresponding boundary site in the adjacent lane. At the entrance of each lane, $+$ and $-$ particles are injected at rates $\alpha^+$ and $\alpha^-$, respectively, provided the entrance site is vacant and the corresponding site on the adjacent lane is either empty or occupied by a particle of same species only. However, if the corresponding entrance site on the adjacent lane is occupied by a particle of the opposite species, entry is prohibited, as illustrated in Fig. 1. Moreover, the particles detach from the exit site of each lane with a rate $\beta$, irrespective of the occupancy state of the corresponding site on the adjacent lane.

We introduce two binary variables, $\tau_i^k$ and $\eta_i^k$, where $k \in \{A, B\}$, to represent the occupancy status of positive and negative particles, respectively, at site $i$ on lane $k$. Specifically, $\tau_i^k$ ($\eta_i^k$) takes the value '1' if a $+$ $(-)$ particle occupies the $i^{\mathrm{th}}$ site of lane $k$, otherwise it is set to '0'. At each time step, a lane site $(i, k)$, where $i \in \{1, 2, \ldots, L\}$, $k \in \{A, B\}$ is selected, and the system evolves according to random sequential update rules. Since particles on lanes A and B do not interact in the bulk, the master equations governing the temporal evolution of occupancy for both particle species within the bulk of lane $k$ can simply be given by

$$\frac{d}{dt} \begin{bmatrix} \langle \tau_i^k \rangle \\ \langle \eta_i^k \rangle \end{bmatrix} = \begin{bmatrix} J_{i-1,i}^{k^+} - J_{i,i+1}^{k^+} \\ J_{i+1,i}^{k^-} - J_{i,i-1}^{k^-} \end{bmatrix}, \tag{4}$$

where $\langle \cdots \rangle$ denotes the statistical average. The quantities $J_{i-1,i}^{k^+}$ and $J_{i+1,i}^{k^-}$ represent the particle currents in lane $k$ due to $+$ and $-$ particle species, respectively, and are expressed as

$$\begin{bmatrix} J_{i-1,i}^{k^+} \\ J_{i+1,i}^{k^-} \end{bmatrix} = \begin{bmatrix} \langle \tau_{i-1}^k (1 - \tau_i^k - \eta_i^k) \rangle + q \langle \tau_{i-1}^k \eta_i^k \rangle \\ \langle \eta_{i+1}^k (1 - \tau_i^k - \eta_i^k) \rangle + q \langle \eta_{i+1}^k \tau_i^k \rangle \end{bmatrix}. \tag{5}$$

The first and the second terms on the right-hand sides of the above equations correspond
to the hopping of a particle to the neighboring vacant site and the interchange of the two
oppositely directed particle species in the appropriate direction, respectively. It is evident
from Eq. (5) that the bulk current equations decouple only when $q = 1$. For simplicity,
we restrict our analysis to this case in the subsequent discussion. Accordingly, the current
expressions given in Eq. (5) simplify to the following forms:

$$
\begin{bmatrix} J_{i-1,i}^{k+} \\ J_{i+1,i}^{k-} \end{bmatrix} = \begin{bmatrix} \langle \tau_{i-1}^k (1 - \tau_i^k) \rangle \\ \langle \eta_{i+1}^k (1 - \eta_i^k) \rangle \end{bmatrix}.
\tag{6}
$$

Clearly, the particle of one species does not distinguish between a hole and a particle of
other species during its attempt to hop to next site. However, due to the presence of
narrow entrances, the boundary behavior differs from that in the bulk. As a result, the
evolution equations governing particle dynamics at the boundary sites $i = 1$ and $i = L$,
on lane $k$ are written as

$$
\frac{d}{dt} \begin{bmatrix} \langle \tau_1^k \rangle \\ \langle \eta_1^k \rangle \end{bmatrix} = \begin{bmatrix} J_{\text{enter}}^{k+} - J_{1,2}^{k+} \\ J_{2,1}^{k-} - J_{\text{exit}}^{k-} \end{bmatrix}, \qquad \frac{d}{dt} \begin{bmatrix} \langle \tau_L^k \rangle \\ \langle \eta_L^k \rangle \end{bmatrix} = \begin{bmatrix} J_{L-1,L}^{k+} - J_{\text{exit}}^{k+} \\ J_{\text{enter}}^{k-} - J_{L,L-1}^{k-} \end{bmatrix},
\tag{7}
$$

where

$$
\begin{bmatrix} J_{\text{enter}}^{A+} \\ J_{\text{enter}}^{B+} \\ J_{\text{enter}}^{A-} \\ J_{\text{enter}}^{B-} \end{bmatrix} = \begin{bmatrix} \alpha^+ \langle (1 - \eta_1^B)(1 - \tau_1^A - \eta_1^A) \rangle \\ \alpha^+ \langle (1 - \eta_1^A)(1 - \tau_1^B - \eta_1^B) \rangle \\ \alpha^- \langle (1 - \tau_L^B)(1 - \tau_L^A - \eta_L^A) \rangle \\ \alpha^- \langle (1 - \tau_L^A)(1 - \tau_L^B - \eta_L^B) \rangle \end{bmatrix}, \quad \begin{bmatrix} J_{\text{exit}}^{A+} \\ J_{\text{exit}}^{B+} \\ J_{\text{exit}}^{A-} \\ J_{\text{exit}}^{B-} \end{bmatrix} = \begin{bmatrix} \beta \langle \tau_L^A \rangle \\ \beta \langle \tau_L^B \rangle \\ \beta \langle \eta_1^A \rangle \\ \beta \langle \eta_1^B \rangle \end{bmatrix}.
\tag{8}
$$

The master equations derived above contain both one-point and two-point correlators,
which renders them analytically intractable in their exact form. To address this com-
plexity, we employ the mean-field approximation, a standard technique in the study of
TASEP models, which simplifies the analysis by neglecting all particle correlations. This
approximation allows the multi-point correlator functions to be expressed as products of
individual one-point correlator functions. In particular, for all $i$ and $j$,

$$
\langle \tau_i^k \tau_j^k \rangle = \langle \tau_i^k \rangle \langle \tau_j^k \rangle, \quad \langle \eta_i^k \eta_j^k \rangle = \langle \eta_i^k \rangle \langle \eta_j^k \rangle.
\tag{9}
$$

To further analyze the master equations in the thermodynamic limit $(L \to \infty)$, the dis-
crete lattice model is reformulated into a coarse-grained continuum description. This
transformation facilitates a more tractable analysis by allowing the lattice spacing to in-
finitesimally small, thereby yielding a continuous representation of the system's dynamics.
In this version, the mean-field densities of both species of particles at site $i$ in lane $k$ are
designated by $\rho_i^k = \langle \tau_i^k \rangle$ and $\sigma_i^k = \langle \eta_i^k \rangle$, respectively. Therefore, the currents correspond
to both the particle species in lane $k$ can be expressed as

$$
\begin{bmatrix} J_{i-1,i}^{k+} \\ J_{i+1,i}^{k-} \end{bmatrix} = \begin{bmatrix} \rho_{i-1}^k \left(1 - \rho_i^k\right) \\ \sigma_{i+1}^k \left(1 - \sigma_i^k\right) \end{bmatrix}.
\tag{10}
$$

The continuum version of proposed system is obtained by coarse-graining the discrete
lattice by introducing a quasicontinuous spatial variable $x = i/L \in [0, 1]$ with the lattice
spacing constant $\epsilon = 1/L$, and rescaling time variable as $t' = t/L$. Expanding the discrete

master equations (4) using a Taylor series and retaining terms up to second order, we obtain

$$\frac{\partial \rho^k}{\partial t'} = \frac{\epsilon}{2}\frac{\partial^2 \rho^k}{\partial x^2} + (2\rho^k - 1)\frac{\partial \rho^k}{\partial x}, \tag{11}$$

$$\frac{\partial \sigma^k}{\partial t'} = \frac{\epsilon}{2}\frac{\partial^2 \sigma^k}{\partial x^2} - (2\sigma^k - 1)\frac{\partial \sigma^k}{\partial x}. \tag{12}$$

Under the assumption of spatial homogeneity, the subscript $i$ is dropped from the density variables. The steady-state densities are obtained by setting the time derivatives in Eqs. (11) and (12) to zero. This leads to a pair of singularly perturbed differential equations which can be expressed as

$$\frac{\epsilon}{2}\frac{\partial^2 \rho^k}{\partial x^2} + (2\rho^k - 1)\frac{\partial \rho^k}{\partial x} = 0, \tag{13}$$

$$\frac{\epsilon}{2}\frac{\partial^2 \sigma^k}{\partial x^2} - (2\sigma^k - 1)\frac{\partial \sigma^k}{\partial x} = 0. \tag{14}$$

When the perturbation parameter $\epsilon \to 0$, the above equations reduce to $(2\rho^k - 1)\frac{\partial \rho^k}{\partial x} = 0$ and $(1 - 2\sigma^k)\frac{\partial \sigma^k}{\partial x} = 0$, which implies $\frac{\partial J^{k^+}}{\partial x} = 0$ and $\frac{\partial J^{k^-}}{\partial x} = 0$. This indicates that the bulk particle currents $J^{k^+}$ and $J^{k^-}$ corresponding to $+$ and $-$ species, respectively, remain constant and are given by

$$\begin{bmatrix} J^{k^+} \\ J^{k^-} \end{bmatrix} = \begin{bmatrix} \rho^k(1 - \rho^k) \\ \sigma^k(1 - \sigma^k) \end{bmatrix}. \tag{15}$$

Likewise, the currents at the boundaries can be expressed as

$$\begin{bmatrix} J_{\text{enter}}^{A^+} \\ J_{\text{enter}}^{B^+} \\ J_{\text{enter}}^{A^-} \\ J_{\text{enter}}^{B^-} \end{bmatrix} = \begin{bmatrix} \alpha^+(1 - \sigma_1^B)(1 - \rho_1^A - \sigma_1^A) \\ \alpha^+(1 - \sigma_1^A)(1 - \rho_1^B - \sigma_1^B) \\ \alpha^-(1 - \rho_L^B)(1 - \rho_L^A - \sigma_L^A) \\ \alpha^-(1 - \rho_L^A)(1 - \rho_L^B - \sigma_L^B) \end{bmatrix}, \quad \begin{bmatrix} J_{\text{exit}}^{A^+} \\ J_{\text{exit}}^{B^+} \\ J_{\text{exit}}^{A^-} \\ J_{\text{exit}}^{B^-} \end{bmatrix} = \begin{bmatrix} \beta\rho_L^A \\ \beta\rho_L^B \\ \beta\sigma_1^A \\ \beta\sigma_1^B \end{bmatrix}. \tag{16}$$

It is evident from the Eqs. (15) and (16), the bulk currents of both species are decoupled while interactions between particle species occurring solely at the system boundaries. These boundary interactions arise from two distinct mechanism of the system: first, from the coupling between the two lanes due to narrow entrances, where the entry of a particle into one lane depends on the occupancy of the corresponding site in the adjacent lane; and second, from the competition between oppositely directed particle species within the same lane, as both species attempt to access the same boundary site for entry or exit.

To quantitatively account for the coupling induced by narrow entrances, we define modified entrance rates $\alpha_{mod}^{k^+}$ and $\alpha_{mod}^{k^-}$ for the $+$ and $-$ particles in lane $k \in \{A, B\}$, that incorporate the influence of inter-lane interactions and are elucidated as

$$\alpha_{mod}^{A^+} = \alpha^+(1 - \sigma_1^B), \quad \alpha_{mod}^{A^-} = \alpha^-(1 - \rho_L^B),$$
$$\alpha_{mod}^{B^+} = \alpha^+(1 - \sigma_1^A), \quad \alpha_{mod}^{B^-} = \alpha^-(1 - \rho_L^A). \tag{17}$$

As a result, the whole system can effectively be interpreted as two single-lane bidirectional TASEP models with modified entrance as $\alpha_{mod}^{k^+}$ and $\alpha_{mod}^{k^-}$ and same exit rate $\beta$ for both $+$ and $-$ particle species.

Now, to quantify the boundary effects resulting from the competition between the two distinct particle species at the entry sites, it is essential to define the effective entrance rates $\alpha_{eff}^{k^+}$ and $\alpha_{eff}^{k^-}$ for the $+$ and $-$ particles, respectively, on a given lane $k$ [28]. These rates are obtained by enforcing current continuity between the bulk and boundary regions of the lane, and are expressed as:

$$
\alpha_{eff}^{k^+} = \frac{\alpha_{mod}^{k^+}(1 - \rho_1^k - \sigma_1^k)}{1 - \rho_1^k} = \frac{J^{k^+}}{\frac{J^{k^+}}{\alpha_{mod}^{k^+}} + \frac{J^{k^-}}{\beta}},
$$

$$
\alpha_{eff}^{k^-} = \frac{\alpha_{mod}^{k^-}(1 - \rho_L^k - \sigma_L^k)}{1 - \sigma_L^k} = \frac{J^{k^-}}{\frac{J^{k^-}}{\alpha_{mod}^{k^-}} + \frac{J^{k^+}}{\beta}}. \tag{18}
$$

Thus, the entire system reduces to four independent single species TASEP lanes, with the boundary parameters $(\alpha_{eff}^{k^+}, \beta)$ and $(\alpha_{eff}^{k^-}, \beta)$ governing the dynamics of the positive and negative particle species, respectively, on lane $k$.

An important factor that influence the determination of the density profile is the total number of particles present in the system. Since the boundaries of both the lanes are connected to two distinct reservoirs, $R^+$ and $R^-$, this global constraint can be incorporated in the continumm limit by approximating the discrete particle conservation condition given in Eq. (1), via a continuous Riemann sum, yielding the following expressions

$$
\mu^+ = r^+ + \frac{1}{2}\int_0^1 \left[\rho^A(x) + \rho^B(x)\right]dx, \quad \mu^- = r^- + \frac{1}{2}\int_0^1 \left[\sigma^A(x) + \sigma^B(x)\right]dx. \tag{19}
$$

Here, $\mu^+ = N^{tot^+}/(2L)$ and $\mu^- = N^{tot^-}/(2L)$ are the filling factors, which indicates the fraction of total $+$ and $-$ particles relative to the total number of sites in the system and $r^+$ and $r^-$ denotes the densities of the reservoirs $R^+$ and $R^-$, respectively, in the continuous framework. Now, our objective is to characterize the stationary properties of the system including phase diagrams, density profiles, particle currents, and possible phase transitions by calculating the effective entrance rates $\alpha_{eff}^{k^\pm}$ for both the particle species by utilizing the Eqs. (15), (16) and (18) along with Eq. (19). To obtain the full density profiles, we proceed by separately calculating the bulk (outer) solution and the boundary layer (inner) solution, which together capture the complete spatial structure of the system.

## 3 Existence of phases

To investigate the effect of coupling the lanes to two distinct particle reservoirs, we analyze its behavior in the $\alpha - \beta$ parameter space by first identifying the possible steady-state phases, followed by a comprehensive study of stationary-state properties such as density profiles, particle currents, and phase transitions. The proposed model under infinite resources exhibits spontaneous symmetry breaking, giving rise to two symmetric and two asymmetric phases, which are classified based on the characteristics of their stationary density profiles, particle currents, and effective entrance rates [57]. To characterize the symmetric (asymmetric) phases in our model, we use the notations: LD (L) for low density phase, HD (H) for high density phase, MC (M) for maximal current and SP (S) for shock phase.

Let us now investigate the possible stationary phases that may emerge in a two-lane bidirectional model with constrained entrances coupled to two distinct finite particle reservoirs. We denote a phase by the notation P-Q/R-S, where P and R represent the phases exhibited by $+$ particles in lanes A and B, respectively, while Q and S indicate the phases

of $-$ particles in the corresponding lanes. In the proposed model, each species in both lanes can exhibit any of the four distinct phases: low density, high density, maximal current, or shock. We now proceed to discuss the symmetric and asymmetric phases that emerge in the system in detail.

## 3.1  Symmetric phases

Here, we focus on the emergence of various symmetric phases and aim to derive the analytical expressions for effective rates, density profiles, and phase boundaries associated with these phases. In these phases, both particle species exhibit identical behavior as well as stationary properties, including effective entrance rates, bulk densities, and currents, in both lanes. In particular, for $k \in \{A, B\}$, the following conditions hold: $\mu^+ = \mu^-$, $\alpha_{mod}^{k^+} = \alpha_{mod}^{k^-}$, $\alpha_{eff}^{k^+} = \alpha_{eff}^{k^-}$, $\rho^k = \sigma^k$ and $J^{k^+} = J^{k^-}$ and in addition from Eq. (19), we have $r^+ = r^-$.

The whole system can be viewed as four independent single species TASEP models, which are coupled solely at the boundaries. Thus, each particle species in both lanes can exhibit one of the four possible phases: LD, HD, MC, or SP. In view of the case under consideration, where both particle species exhibit identical behavior and stationary properties, the possible configurations reduce to four symmetric combinations: LD-LD/LD-LD, HD-HD/HD-HD, MC-MC/MC-MC and SP-SP/SP-SP. However, among these four symmetric phases, only two are physically realizable namely LD–LD/LD–LD and MC–MC/MC–MC. The existence of HD-HD/HD-HD phase is ruled out since it would require the total particle density to exceed unity in a given lane, which is not physically permissible. On the other hand, the existence of SP-SP/SP-SP phase requires the condition $\alpha_{eff}^{k^\pm} = \beta$ to be satisfied. However this condition has no feasible solution for any value of common filling factor $\mu^\pm$. Now, we will proceed to examine each of the remaining phases in detail.

1. *Low density phase* (LD-LD/LD-LD):

In this phase, both the species of particles in each lane are in low density with bulk densities equal to $\alpha_{eff}^{k^+}$ and $\alpha_{eff}^{k^-}$ and the corresponding current for both particle species are equal throughout the lanes, i.e., $J_{enter}^{k^+} = J_{exit}^{k^+} = J_{enter}^{k^-} = J_{exit}^{k^-} = J^{k^+} = J^{k^-}$. Now by utilizing the condition of current continuity, we have $\alpha_{eff}^{A^+}(1 - \alpha_{eff}^{A^+}) = J_{exit}^{B^-}$ and further substituting this into Eqs. (16) and (17), the modified entry rate for $+$ particles in lane A becomes

$$\alpha_{mod}^{A^+} = \alpha \, \frac{r^\pm}{\mu^\pm} \left( 1 - \frac{\alpha_{eff}^{A^+}(1 - \alpha_{eff}^{A^+})}{\beta} \right). \tag{20}$$

By employing Eq. (18) the stationary density of $+$ particle species in lane A for this phase is given by,

$$\alpha_{eff}^{A^+} = \frac{\alpha_{mod}^{A^+} \beta}{\alpha_{mod}^{A^+} + \beta}. \tag{21}$$

Note that $\alpha_{mod}^{A^+}$ is governed by the reservoir density $r^\pm$ and the filling factor $\mu^\pm$. By utilizing Eq. (19), the reservoir density can be simplified as

$$r^\pm = \mu^\pm - \alpha_{eff}^{A^+}. \tag{22}$$

In this symmetric case, where $\alpha_{eff}^{k^+} = \alpha_{eff}^{k^-}$ for $k \in \{A, B\}$, the existence of this phase necessitates that the effective entrance rates to be less than 0.5 and $\beta$, i.e.

$$\alpha_{eff}^{k^\pm} < \min\{0.5, \; \beta\}. \tag{23}$$

336  2. *Maximal current Phase* (MC-MC/MC-MC):

337     This phase arises when both particle species of each lane individually reside in the max-
338     imal current phase with bulk densities 0.5 and bulk particle currents $J^{k^+} = J^{k^-} = 0.25$.
339     From Eq. (16), this yields the boundary density $\sigma_1^B = 1/(4\beta)$. Further by employing
340     Eqs. (17)- (19) the effective rates are given by

$$\alpha_{eff}^{k^\pm} = \frac{\alpha \, r^\pm \beta (4\beta - 1)}{\alpha \, r^\pm (4\beta - 1) + 4\beta^2 \mu^\pm}, \tag{24}$$

341     with reservoir density $r^\pm = \mu^\pm - 0.5$. The existential condition for this phase is char-
342     acterized by:

$$\min\{\alpha_{eff}^{k^\pm}, \beta\} > 0.5. \tag{25}$$

### 343  3.2  Asymmetric phases

344  The symmetry of the system is disrupted by localized interactions between the two particle
345  species only at the boundaries, resulting in spontaneous symmetry breaking (SSB) even
346  when the global condition $\mu^+ = \mu^-$ is satisfied. Clearly, for the case $\mu^+ \neq \mu^-$, the system
347  solely exhibits asymmetric phases. Even if both the particle species reside in the same
348  phase within each lane, their stationary properties remain distinct due to the inherent
349  asymmetry in reservoir conditions. In particular, the bulk densities of the positive and
350  negative particles are unequal, i.e., $\rho^k \neq \sigma^k$ for $k \in \{A, B\}$, which leads to $\alpha_{eff}^{k^+} \neq \alpha_{eff}^{k^-}$.
351  Here, each particle species can exhibit any of the four phases in each lane: low density
352  (L), high density (H), shock (S), or maximal current (M). As a result, the total number
353  of possible phases for the system becomes $4^4 = 256$.

354     Considering an individual lane, it can exhibit a total of $4^2 = 16$ possible phases.
355  However, only 5 of these phases are physically admissible, while the remaining 11 are
356  excluded based on the following criteria: (i) the combined densities of the two species in
357  a lane must satisfy the constraint $(\rho^k + \sigma^k \leq 1)$ (e.g., M-H, H-H, etc.); (ii) symmetry
358  between the + and − species renders some phases equivalent (e.g., S-L and L-S, M-L and
359  L-M etc.). Now, extending this analysis to the two-lane system leads to $5^2 = 25$ possible
360  phase combinations out of which 20 phases are eliminated based on the above criteria
361  along with two additional considerations: (i) certain phase combinations are incompatible
362  with each other and cannot coexist in the stationary state; (e.g. M-M/H-L and L-L/M-L
363  etc.); (ii) numerical investigations corroborated by Monte Carlo simulations, confirm the
364  absence of several theoretically feasible phases (e.g. L-L/H-L and M-M/L-L etc.). To
365  determine the effective entrance rates for the admissible phases, we first calculate the
366  reservoir densities by enforcing particle number conservation for each particle species.
367  These expressions will subsequently be used to derive the phase boundaries, identify the
368  shock positions, and calculate the particle densities. In the following, we examine in detail
369  the necessary conditions for the emergence of the possible asymmetric phases.

370  1. *High-low density phase* (H-L/H-L): In this phase, we assume that the + particles reside
371     in the high density regime with bulk density greater than 0.5, while the − particles
372     exhibit the low density phase in the bulk of both lanes. Consequently, from Eq. (15),
373     we have

$$J^{k^+} = \beta(1 - \beta), \quad J^{k^-} = \alpha_{eff}^{k^-}(1 - \alpha_{eff}^{k^-}). \tag{26}$$

374  Utilizing Eqs. (16), (17), and (26), we have

$$\alpha_{mod}^{A^-} = \alpha_{mod}^{B^-} = \alpha^- \beta, \tag{27}$$

and

$$\alpha_{mod}^{A^+} = \alpha^+ \left( 1 - \frac{\alpha_{eff}^{B^-}(1 - \alpha_{eff}^{B^-})}{\beta} \right), \quad \alpha_{mod}^{B^+} = \alpha^+ \left( 1 - \frac{\alpha_{eff}^{A^-}(1 - \alpha_{eff}^{A^-})}{\beta} \right). \tag{28}$$

Since $\alpha_{mod}^{A^-} = \alpha_{mod}^{B^-}$, from Eqs. (18) and (26), it follows that $\alpha_{eff}^{A^-} = \alpha_{eff}^{B^-}$ and on further investigation, it gives $\alpha_{eff}^{A^+} = \alpha_{eff}^{B^+}$. Now, on solving the above equations the effective entry rates are given by

$$\alpha_{eff}^{A^-} = \frac{\mu^- + \alpha\beta(\beta + \mu^-) - \sqrt{(\mu^- + \alpha\beta(\beta + \mu^-))^2 - 4\alpha\beta^2\mu^-(\alpha\beta + \mu^-)}}{2(\alpha\beta + \mu^-)},$$

$$\alpha_{eff}^{A^+} = -\frac{\alpha(\beta - 1)\beta^2 \, r^+ \left( \beta - \alpha_{eff}^{A^-} + (\alpha_{eff}^{A^-})^2 \right)}{\beta^3\mu^+(1 - \beta) + \alpha \, r^+ \, \alpha_{eff}^{A^-}(1 - \alpha_{eff}^{A^-})\left[\beta - \alpha_{eff}^{A^-}(1 - \alpha_{eff}^{A^-})\right]}. \tag{29}$$

Utilizing the particle number conservation given by Eq. (19) yields the following expressions for reservoir densities,

$$r^+ = \mu^+ - (1 - \beta), \quad r^- = \mu^- - \alpha_{eff}^{A^-}. \tag{30}$$

Since $\alpha_{eff}^{A^+} = \alpha_{eff}^{B^+}$ and $\alpha_{eff}^{A^-} = \alpha_{eff}^{B^-}$, the existential condition for this phase satisfies

$$\min\{\alpha_{eff}^{k^+}, 0.5\} > \beta, \quad \alpha_{eff}^{k^-} < \min\{\beta, 0.5\}, \tag{31}$$

for $k \in \{A, B\}$ along with the filling factors satisfying $\mu^+ \geq \mu^-$ and $\mu^+ > 0.5$. Furthermore, the condition $\alpha_{eff}^{k^+} > \beta$ is fulfilled only when $\mu^- < 0.5 < \mu^+$.

2. *Shock-low density phase* (S-L/S-L): In this phase, the $+$ particles exhibit a shock phase, while the $-$ particles remain in the low density across both the lanes. The existence of this phase is ensured when the boundary-regulating parameters fulfill the following constraints:

$$J^{k^+} = \beta(1 - \beta) = \alpha_{eff}^{k^+}(1 - \alpha_{eff}^{k^+}),$$
$$J^{k^-} = \alpha_{eff}^{k^-}(1 - \alpha_{eff}^{k^-}). \tag{32}$$

The effective entrance rates for the $+$ and $-$ particle species can be retrieved from the Eq. (29) as

$$\alpha_{eff}^{A^-} = \frac{\mu^- + \alpha\beta(\beta + \mu^-) - \sqrt{(\mu^- + \alpha\beta(\beta + \mu^-))^2 - 4\alpha\beta^2\mu^-(\alpha\beta + \mu^-)}}{2(\alpha\beta + \mu^-)},$$

$$\alpha_{eff}^{A^+} = -\frac{\alpha(\beta - 1)\beta^2 \, r^+ \left( \beta - \alpha_{eff}^{A^-} + (\alpha_{eff}^{A^-})^2 \right)}{\beta^3\mu^+(1 - \beta) + \alpha \, r^+ \, \alpha_{eff}^{A^-}(1 - \alpha_{eff}^{A^-})\left[\beta - \alpha_{eff}^{A^-}(1 - \alpha_{eff}^{A^-})\right]}. \tag{33}$$

The existence of this phase requires $\alpha_{eff}^{A^+} = \beta$, which gives the reservoir density for $+$ particle species as

$$r^+ = \frac{(-1 + \beta)\beta^3\mu^+}{\alpha\left[\beta^3 + \alpha_{eff}^{A^-}(1 - \alpha_{eff}^{A^-})(2\beta - \alpha_{eff}^{A^-}(1 - \alpha_{eff}^{A^-})) + \beta^2\left((\alpha_{eff}^{A^-})^2 - \alpha_{eff}^{A^-} - 1\right)\right]}. \tag{34}$$

Since $\alpha_{eff}^{A^+} = \alpha_{eff}^{B^+}$ and $\alpha_{eff}^{A^-} = \alpha_{eff}^{B^-}$, by utilizing the Eq. (19)

$$\int_0^1 \rho^A dx = \int_0^{x_w} \alpha_{eff}^{A^+} dx + \int_{x_w}^1 (1-\beta) dx. \tag{35}$$

Here, $x_w$ denotes the shock position in the density profile of the + particles. This position can be determined using Eqs. (19), (34), and (35), and is given by

$$x_w = \frac{1}{1-2\beta} \left( 1 - \beta - \mu^+ + r^+ \right). \tag{36}$$

The reservoir density for $-$ particle species satisfies $r^- = \mu^- - \alpha_{eff}^{A^-}$. Finally, the existence of the S–L/S-L phase requires that the boundary parameters satisfy the following set of conditions for $k \in \{A, B\}$:

$$0 \le x_w \le 1, \quad \alpha_{eff}^{k^-} < \min\{\beta, 0.5\}. \tag{37}$$

3. *Maximal-low density phase* (M-L/M-L): Here, the + particles exhibit the maximal current phase, characterized by a bulk density of 0.5, whereas the $-$ particles remain in the low density regime with an average density less than 0.5 across both lanes. The corresponding bulk particle currents are given by

$$J^{k^+} = 0.25, \quad J^{k^-} = \alpha_{eff}^{k^-}(1 - \alpha_{eff}^{k^-}). \tag{38}$$

By utilizing the Eqs. (16), (17), (18) and (38) the effective entrance rates for both particle species can be expressed as follows:

$$
\begin{aligned}
\alpha_{eff}^{A^-} =& \ \frac{1}{8\beta\left(\alpha(-1+4\beta)+4\beta\mu^-\right)} \Bigg( 16\beta^2\mu^- + \alpha(-1+4\beta)\big(-1+4\beta(1+\mu^-)\big) \\
&- \sqrt{\big(16\beta^2\mu^- + \alpha(-1+4\beta)(-1+4\beta(1+\mu^-))\big)^2 - 16\alpha(1-4\beta)^2\beta\mu^-\left(\alpha(-1+4\beta)+4\beta\mu^-\right)} \Bigg), \\
\alpha_{eff}^{A^+} =& \ \frac{\alpha\beta r^+ \left(\beta - \alpha_{eff}^{A^-} + (\alpha_{eff}^{A^-})^2\right)}{\beta^2\mu^+ - 4\alpha r^+(\alpha_{eff}^{A^-})^2(1-\alpha_{eff}^{A^-})^2 - 4\alpha\beta r^+\alpha_{eff}^{A^-}(1-\alpha_{eff}^{A^-})}
\end{aligned}
\tag{39}
$$

Since it is observed that $\alpha_{eff}^{A^+} = \alpha_{eff}^{B^+}$ and $\alpha_{eff}^{A^-} = \alpha_{eff}^{B^-}$, the reservoir density for both the particle species can be determined by using Eq. (19) as

$$r^+ = \mu^+ - 0.5, \quad r^- = \mu^- - \alpha_{eff}^{A^-}.$$

Using the boundary parameter expressions derived above, the criteria for existence of this phase are formulated as

$$\alpha_{eff}^{k^-} < 0.5 < \min\{\alpha_{eff}^{k^+}, \beta\}, \quad \mu^+ > \max\{\mu^-, 0.5\}. \tag{40}$$

4. *Maximal-maximal density phase* (M-M/M-M): All stationary properties of this phase resemble those of the MC-MC/MC-MC phase, except that $\alpha_{eff}^{k^+} \neq \alpha_{eff}^{k^-}$, a consequence of asymmetry in filling factors $\mu^+ \neq \mu^-$. In this phase, the bulk density of both particle species in each lane remains fixed at 0.5 with currents $J^{k^+} = J^{k^-} = 0.25$. However, the boundary densities are distinct, such that $\rho^k(0) \neq \sigma^k(0)$ and $\rho^k(1) \neq \sigma^k(1)$. The effective entrance rates for both positive and negative particle species in lane $k \in \{A, B\}$ are given by

$$\alpha_{eff}^{k^j} = \frac{\alpha r^j \beta(4\beta - 1)}{\alpha r^j(4\beta - 1) + 4\beta^2\mu^j}. \tag{41}$$

The reservoir densities are obtained by using particle conservation law given in Eq. (19), yielding $r^j = \mu^j - 0.5$ for $j \in \{+, -\}$. The existential condition of this phase is given by

$$\min\{\alpha_{eff}^{k^+}, \alpha_{eff}^{k^-}, \beta, \ \mu^+, \mu^-\} > 0.5. \tag{42}$$

5. *Low-low density phase* (L-L/L-L): This phase emerges when the densities of both particle species are entry dominated with values remaining below 0.5, while the effective entry rates differ, i.e., $\alpha_{eff}^{k^+} \neq \alpha_{eff}^{k^-}$. The corresponding bulk current for each particle species in lane $k \in \{A, B\}$ are given by

$$J^{k^+} = \alpha_{eff}^{k^+}(1 - \alpha_{eff}^{k^+}), \quad J^{k^-} = \alpha_{eff}^{k^-}(1 - \alpha_{eff}^{k^-}). \tag{43}$$

Utilizing the particle number conservation along with the relation $\rho^k = \alpha_{eff}^{k^+}$ and $\sigma^k = \alpha_{eff}^{k^-}$, we have the expressions for reservoir densities as

$$r^+ = \mu^+ - \alpha_{eff}^{k^+}, \quad r^- = \mu^- - \alpha_{eff}^{k^-}.$$

The effective entrance rates for both particle species are determined by solving Eqs. (16), (17), and (43). The existential condition for this phase is given by

$$\max\{\alpha_{eff}^{k^+}, \alpha_{eff}^{k^-}\} < \min\{\beta, 0.5\}. \tag{44}$$

In the case of symmetric filling factors, all of the above expressions simplify by substituting $\mu^+ = \mu^-$.

# 4  Results and discussion

In this section, we investigate the influence of coupling the system to two finite particle reservoirs on its steady-state properties. Our primary objective is to construct the stationary-state phase diagrams in the $\alpha$–$\beta$ parameter space for specific values of the reservoir filling factors $\mu^+$ and $\mu^-$. The values of the filling factors are chosen such that they capture all possible topological transitions in the system, including symmetry breaking phenomena for $\mu^+ = \mu^-$ as well as the appearance of additional asymmetric phases. The theoretical outcomes obtained in the preceding section are validated through a stochastic Monte Carlo simulation based on Gillespie algorithm on a system of size $L = 1500$, evolved over $L \times 10^{11}$ time steps (see Appendix A for detail). To eliminate the effects of transient dynamics arising from different initial conditions, the initial 5% of the total simulation time steps are discarded, ensuring that the system attains a steady-state. Additionally, the numerical scheme described in Appendix B serves as an efficient alternative for computing the analytical results. Now, the subsequent analysis is categorized into two distinct cases: (i) symmetric filling factor ($\mu^+ = \mu^-$) and (ii) asymmetric filling factor ($\mu^+ \neq \mu^-$).

## 4.1  Symmetric filling factor: $\mu^+ = \mu^-$

When the filling factors for both particle species are equal, i.e., $\mu^+ = \mu^-$, the total number of $+$ and $-$ particles in the system remain same. For a less number of particles in the system ($\mu^{\pm} = 0.1$), phase diagram exhibits three phases: one symmetric phase, i.e., LD-LD/LD-LD and two asymmetric phases namely S-L/S-L and L-L/L-L. Despite the inherent

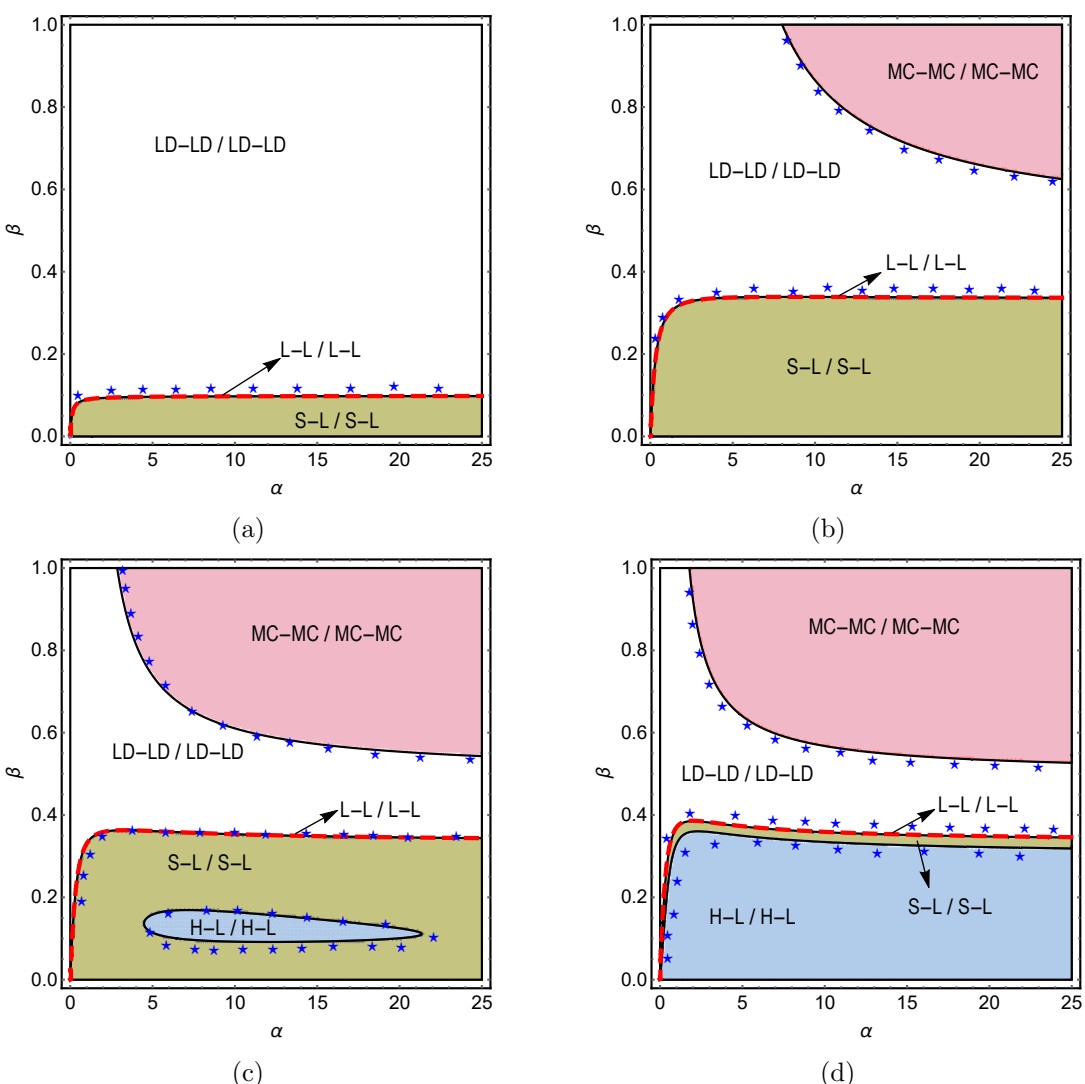

Figure 2: Phase diagrams corresponding to different values of $\mu^{\pm}$: (a) 0.1, (b) 0.6, (c) 0.94 and (d) 2. In each diagram, L-L/L-L phase (red line) remains confined to a curve, serving as a boundary that separates the S-L/S-L and LD-LD/LD-LD phases. Solid lines represent theoretical predictions, while blue symbols denote simulation results for a system of size $L = 1500$.

symmetry between the $+$ and $-$ particles, these two asymmetric phases are still observed [see Fig. 2a]. In these phases, the stationary properties of the $+$ and $-$ particles differ within each lane. This demonstrates that spontaneous symmetry breaking persists even when the total number of particles of each species is significantly low. The asymmetric L-L/L-L phase remains confined to a curve forming a phase boundary separating LD-LD/LD-LD and S-L/S-L phase regions. No qualitative changes are observed in the phase diagram up to $\mu^{\pm} = 0.5$, apart from the gradual expansion of the S-L/S-L phase and a contraction of the LD-LD/LD-LD region.

As the filling factor $\mu^{\pm}$ increases beyond first critical value $\mu_{c_1} = 0.5$, an additional symmetric phase, MC-MC/MC-MC, appears in the phase diagram alongside the previous existing phases, as shown in Fig. 2b. The emergence of symmetric maximal current phase after $\mu^{\pm} = \mu_{c_1}$ is affirmed by the Eq. (25). At this stage, there are sufficient particles in the system to sustain MC-MC/MC-MC phase. No additional symmetric phases emerge

in the system for values of $\mu^\pm$ greater than $\mu_{c_1}$. When $\mu^\pm$ exceeds second critical value $\mu_{c_2} \approx 0.94$, a new asymmetric phase, H-L/H-L, appears in the phase diagram, entirely enclosed by S-L/S-L phase, as illustrated in Fig. 2c. The critical point $\mu_{c_2}$ can be derived from the condition specified in Eq. (31). It is evident that this phase cannot exist for $\mu^\pm \leq \mu_{c_1}$ due to the insufficient number of particles required to sustain it in the system. As $\mu^\pm$, increases further, the region of H-L/H-L phase expands while that of S-L/S-L phase shrinks [see Fig. 2d]. In the limit $\mu^\pm \to \infty$, the S-L/S-L phase completely disappears, resulting in both qualitative and quantitative changes in the phase diagram, consistent with the case of infinite particle reservoirs [57]. Clearly, the number of observed phases in phase diagrams displays a non-monotonic trend as $\mu^\pm$ increases.

## 4.2   Asymmetric filling factor: $\mu^+ \neq \mu^-$

Let us now consider the scenario where the filling factors of the two particle species are unequal. In this regime, even if both species exhibit the same phase, their stationary densities differ, and the notion of spontaneous symmetry breaking no longer prevails. The asymmetry imposed by distinct filling factors ensures that the system exhibits only asymmetric phases. Without loss of generality, we focus on the case $\mu^+ > \mu^-$ and analyze the stationary properties under two scenarios to capture all possible topological changes: (i) fixing small, intermediate, and large values of $\mu^-$ while varying $\mu^+$ and (ii) fixing the value of $\mu^+$ and varying $\mu^-$.

### 4.2.1   Impact of $\mu^+$

Here, we focus on investigating the structural evolution of the phase diagram by fixing the filling factor $\mu^-$ and varying the filling factor $\mu^+$. In the previous section, where both filling factors were identical, significant topological changes in the phase diagrams were observed at the critical values $\mu_{c_1}$ and $\mu_{c_2}$. Based on these critical values, we consider the following scenarios for the present analysis: $0 \leq \mu^- \leq \mu_{c_1}$, $\mu_{c_1} \leq \mu^- \leq \mu_{c_2}$.

- $\mu^- < \mu_{c_1}$:

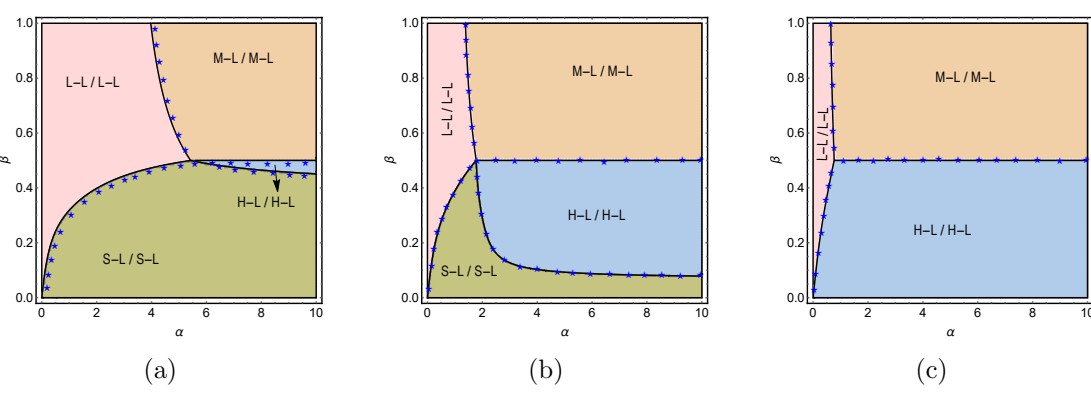

Figure 3: Phase diagrams for (a) $\mu^+ = 0.6$, (b) $\mu^+ = 0.94$, and (c) $\mu^+ = 20$ with $\mu^- = 0.1$. Solid lines represent theoretical predictions, while blue symbols denote simulation results for a system of size $L = 1500$.

In this case, the structure of the phase diagram is significantly influenced by variations in the filling factor $\mu^+$, as shown in Fig. 3. As discussed in Sec. 4.1 for equal filling factors, three distinct phases are observed [see Fig. 2a]. However, when the filling factor become unequal ($\mu^+ \neq \mu^-$), the symmetric LD-LD/LD-LD phase disappears, as

the condition of equal stationary densities between the two species is no longer satisfied. This phase is replaced by the asymmetric L-L/L-L phase, in which both species display low density phases, with unequal stationary densities. It is evident from Fig. 3a that as $\mu^+$ increases beyond $\mu_{c_1}$, two new asymmetric phases, M-L/M-L and H-L/H-L, appear in the phase diagram. Upon further increasing $\mu^+$, the region corresponding to S-L/S-L phase shrinks, while the M-L/M-L and H-L/H-L phase regions expand progressively. This expansion reflects the fact that, with increasing $\mu^+$, the entry rate of $+$ particles increases, while the entry rate of $-$ particles remains unchanged. In the limit $\mu^+ \to \infty$, S-L/S-L phase vanishes completely from the phase diagram, and the H-L/H-L along with M-L/M-L occupies the majority of the phase space, as depicted in Fig. 3c. It can be noted that the number of distinct phases observed with respect to $\mu^+$ depicts a non-monotonic trend.

- $\mu_{c_1} \leq \mu^- \leq \mu_{c_2}$:

For a fixed filling factor of $\mu^- = 0.8$, the phase diagram corresponding to varying $\mu^+$ are presented in Fig. 4. As discussed in Sec. 4.1 for $\mu^+ = \mu^- < \mu_{c_2}$, the phase diagram exhibits four distinct phases [see Fig. 2b], including two symmetric ones: LD-LD/LD-LD and MC-MC/MC-MC. These symmetric phases are highly sensitive to variations in $\mu^+$ and are replaced by the asymmetric L–L/L–L and M–M/M–M phases when $\mu^+ \neq \mu^-$.

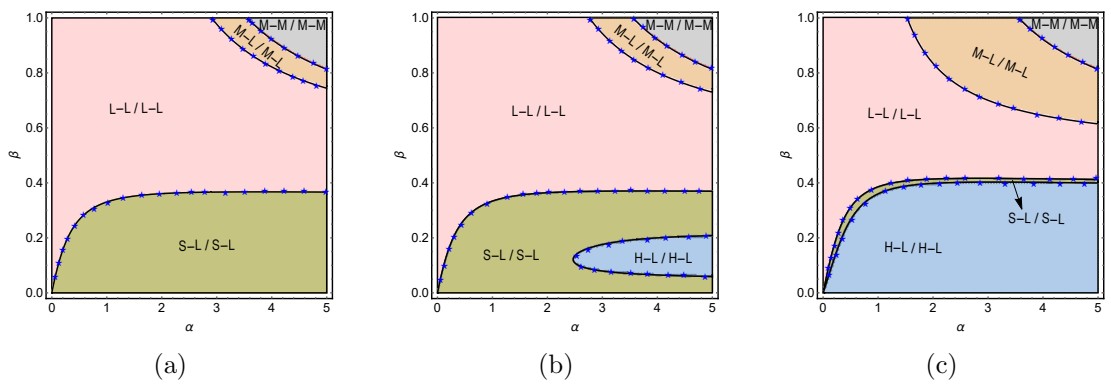

Figure 4: Phase diagrams for (a) $\mu^+ = 0.92$, (b) $\mu^+ = 0.96$, and (c) $\mu^+ = 2.5$ with $\mu^- = 0.8$. Solid lines denotes the theoretical findings and blue symbols correspond to simulation results for a system of size $L = 1500$.

At $\mu_{c_1} \leq \mu^- < \mu^+$, the asymmetric maximal–low density (M–L/M–L) phase appears in the phase diagram [see Fig. 4a]. It is noteworthy to emphasize that, while the bulk characteristics of both $+$ and $-$ particles are identical in the MC-MC/MC-MC and M–M/M-M phases, the particle densities at the boundaries differ implying that $\rho(0) \neq \sigma(1)$ and $\rho(1) \neq \sigma(0)$. This discrepancy arises due to unequal effective entry rates for the two species in the M-M/M-M phase. When $\mu^+$ increases beyond $\mu_{c_2}$, the asymmetric H-L/H-L phase emerges in the phase diagram alongside S-L/S-L phase. Figure 4b corresponds to a scenario where the system exhibits the maximum number of distinct steady-state phases. Further increase in $\mu^+$, leads to an expansion of the M–L/M–L and H–L/H–L regions, while the L–L/L–L and S–L/S–L regions progressively diminish [see Fig. 4c]. In the limit, $\mu^+ \to \infty$, the S-L/S-L phase eventually vanishes from the phase diagram. Notably, the number of distinct steady-state phases exhibits a non-monotonic trend: starting from four, increasing to five as $\mu^+$ rises, and eventually reducing to four again.

### 4.2.2   Impact of $\mu^-$

We now turn our focus on the stationary properties of the system when $\mu^+$ is fixed and $\mu^-$ is varied, under the condition $\mu^+ > \mu^-$. For $\mu^+ < \mu_{c_1}$, the phase diagram exhibits two asymmetric phases: L-L/L-L and S-L/S-L. As observed in Fig. 5a, for $\mu^- < \mu_{c_1} < \mu^+$, the system exhibits four distinct asymmetric phases, namely H-L/H-L, S-L/S-L, L-L/L-L, and M-L/M-L. As $\mu^-$ attains $\mu_{c_1}$, H-L/H-L phase vanishes, accompanied by an expansion of the L-L/L-L phase region and a contraction of the S-L/S-L and M-L/M-L phase regions [see Fig. 5b]. This reflects the fact that with an increase in $\mu^-$ value, a sufficient number of $-$ particles become available in the system, thereby slowing the movement of $+$ particles. As the value of $\mu^-$ exceeds $\mu_{c_1}$, M-M/M-M phase emerges in the phase diagram, as illustrated in Fig. 5c. Notably, for all $\mu^+ > \mu^-$, the density of $+$ particles consistently remains higher than that of $-$ particles in both lanes. One of the significant consequences of coupling two distinct particle reservoirs to the system is the emergence of the M-L/M-L phase in the phase diagram, which has not been observed in the previous studies [28, 57].

In the preceding analysis, our focus was restricted to the case $\mu^+ > \mu^-$. However, the corresponding phase structure, density profiles, and phase diagrams for the case $\mu^+ < \mu^-$ can be derived from the results obtained for $\mu^+ > \mu^-$ by applying the following transformation:

$$
\begin{aligned}
\rho^k &\leftrightarrow \sigma^k, \\
\mu^+ &\leftrightarrow \mu^-, \\
\text{P-Q/R-S} &\leftrightarrow \text{Q-P/S-R.}
\end{aligned}
\tag{45}
$$

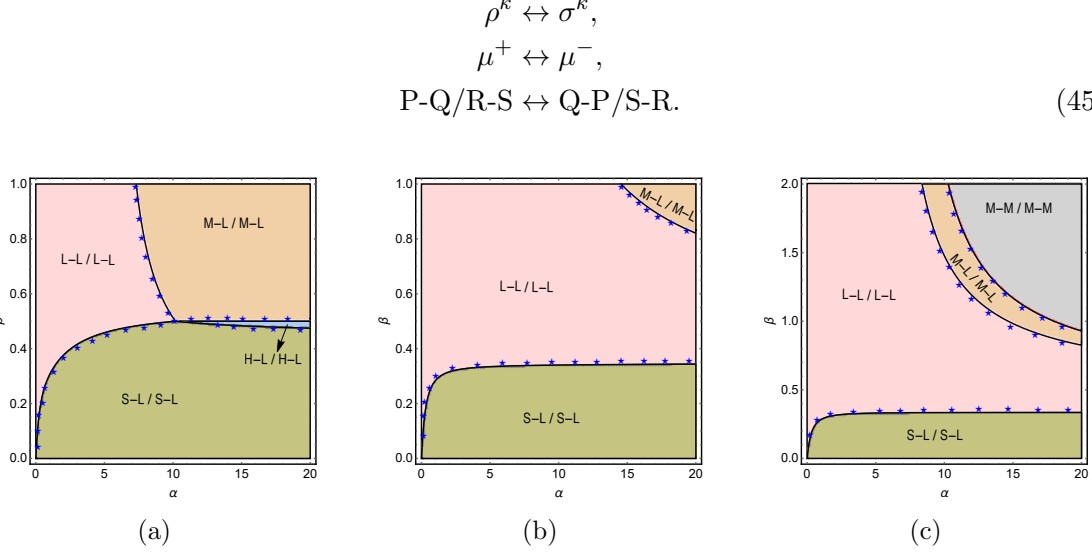

Figure 5: Phase diagrams for (a) $\mu^- = 0.1$ , (b) $\mu^- = 0.5$ and (c) $\mu^- = 0.54$ with $\mu^+ = 0.55$. Solid lines denotes the theoretical findings, while the blue symbols represent the simulation results for a system size of $L = 1500$.

## 4.3   Spontaneous symmetry breaking phenomenon (SSB)

A distinctive feature of bidirectional systems, whether implemented on a single lane or in interacting multiple lanes, is the occurrence of SSB phenomenon [19, 57]. To explore the theoretically observed SSB phenomenon in our system, we perform Monte Carlo simulations based on Gillespie algorithm and construct particle density histograms by monitoring the instantaneous particle densities $\rho^k$ and $\sigma^k$ of the positive and negative particles, respectively, in lane $k$, where $k \in \{A, B\}$. In these simulations, the system size is set to $L = 1500$. To ensure steady-state behavior, the initial $10^{10}$ time steps are discarded, and data is subsequently collected over $9 \times 10^{10}$ time steps. When analyzing a particle density histogram, a phase is classified as symmetric phase if the peaks satisfy $\rho^k = \sigma^k$;

otherwise, it is designated as an asymmetric phase. Figure. 6 displays the particle density histograms for various phases observed in the system. It is evident from the Fig. 6a and 6b that the peak occurs along the diagonal, indicating the presence of symmetric phases. In particular, for LD-LD/LD-LD phase at $(\alpha, \beta, \mu^{\pm}) = (0.9, 0.8, 0.6)$, the particle densities satisfy $\rho^k = \sigma^k < 1/2$, whereas for the MC-MC/MC-MC phase corresponding to $(\alpha, \beta, \mu^{\pm}) = (10, 0.8, 2)$, the peak density attain $\rho^k = \sigma^k = 0.5$.

Figures. 6c- 6e, display off-diagonal peaks, signifying the emergence of asymmetric phases. These phases are obtained under conditions with symmetric filling factor for parameter sets $(\alpha, \beta, \mu^{\pm}) = (0.9, 0.22, 0.6), (0.8, 0.15, 0.94)$ and $(1.5, 0.17, 2)$, correspond to the L-L/L-L, S-L/S-L, and H-L/H-L phases, respectively, each demonstrating spontaneous symmetry breaking. Specifically, for the L-L/L-L phase shown in Fig. 6c, the peak occurs at $\rho^k < \sigma^k < 0.5$, indicating that the densities of positive and negative particles become unequal. For the S-L/S-L phase, Fig. 6d exhibits a peak at $0 < \rho^k < 1$ and $\sigma^k < 1/2$. Similarly, in the H-L/H-L phase depicted in Fig. 6e, the peak is located at $\rho^k > 0.5$ and $\sigma^k < 0.5$, consistent with the expected asymmetry. Notably, in the asymmetric maximal- maximal density (M-M/M-M) phase, the peak remains at $\rho^k = \sigma^k = 0.5$, similar to the symmetric MC-MC/MC-MC phase, but the boundary densities differ due to the asymmetry in the filling factors.

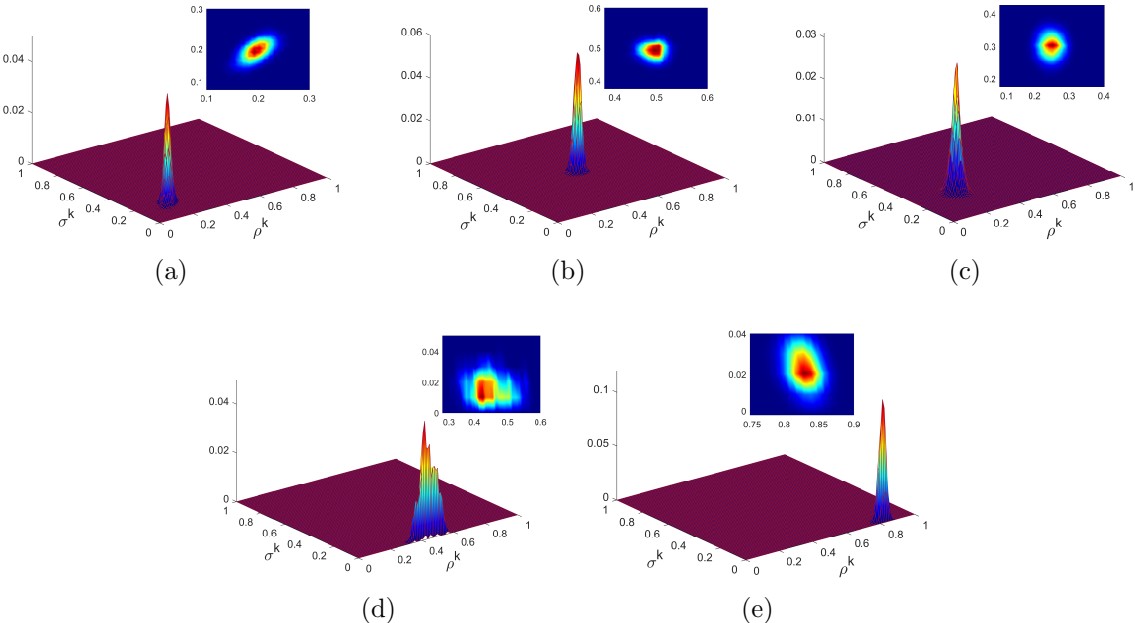

Figure 6: Particle density histograms plotted through Monte Carlo simulations for the following phases: (a) LD-LD/LD-LD, (b) MC-MC/MC-MC, (c) L-L/L-L, (d) S-L/S-L, and (e) H-L/H-L phases.

The SSB phenomenon can also be characterized by examining the behavior of the particle currents for the two species in both the lanes, along with identifying associated phase transitions. To further explore this phenomenon, we examine the variation of the bulk particle currents, $J^{k^+}$ and $J^{k^-}$, as a function of the exit rate $\beta$ while keeping the entry rate fixed at $\alpha = 10$ and the filling factor at $\mu^{\pm} = 0.94$; as depicted in Fig. 7. It is observed that, at a critical value of $\beta \approx 0.353$, both $J^{k^+}$ and $J^{k^-}$ undergo a sudden discontinuity, beyond which they become identical. To further investigate this behavior, we plot the average current, $(J^{k^+} + J^{k^-})/2$, and the absolute current difference, $|J^{k^+} - J^{k^-}|$, as functions of exit rate $\beta$, while keeping other parameters fixed. It is noteworthy that, the average particle current also exhibits a notable discontinuity at the same critical value.

In addition, the behavior of $|J^{k^+} - J^{k^-}|$ undergoes a transition at the same critical point, beyond which it remains constant at zero, confirming the emergence of a symmetric phase. These abrupt changes in the current profiles signify a transition between asymmetric and symmetric phases. Note that, phase diagram for $\mu^\pm = 0.94$, shown in Fig. 2c, illustrates a sequence of phase transition from asymmetric to symmetric phases, transitioning from S-L/S-L $\rightarrow$ H-L/H-L $\rightarrow$ S-L/S-L $\rightarrow$ L-L/L-L $\rightarrow$ LD-LD/LD-LD $\rightarrow$ MC-MC/MC-MC as the exit parameter $\beta$ varies. Additionally, upon analyzing the currents for both particle species, it has been observed that one particle species consistently dominates the transport, exhibiting a higher current than the other. In our case, the $+$ species maintains a current that is always greater than or equal to that of the $-$ species across both lanes. The conditions for the existence of the H-L/H-L and S-L/S-L phases, as derived in Eqs. (31) and (37), demonstrate that the effective entrance rates for the negative particle species are always lower than their corresponding exit rates. This imbalance implies a discontinuous change in the current profiles across the phase boundary, indicating that the transition from asymmetric to symmetric phases is of first-order.

As discussed above, SSB is observed between the two particle species within each lane. However, when examining a single species across both lanes, the stationary properties remain identical as demonstrated in Sec. 3, indicating the absence of SSB in that context. This suggests that the two-lane system with narrow entrances effectively behaves as two independent single-lane TASEPs for each species. Building on this observation, one can infer that an n-lane bidirectional system of identical TASEPs, coupled through narrow entrances, can be reduced into n decoupled identical single-lane TASEPs.

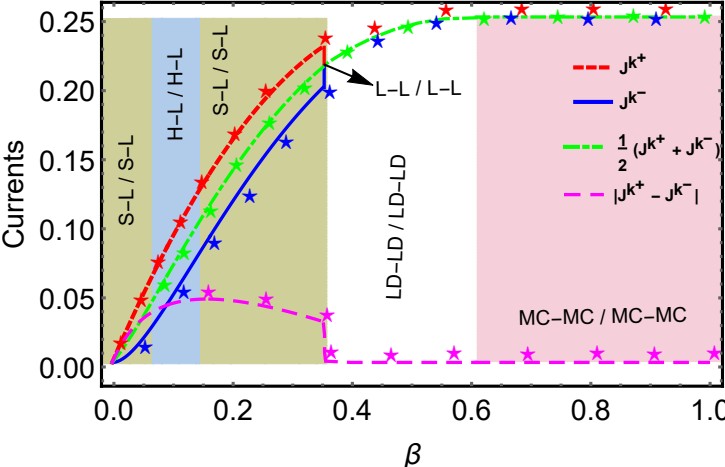

Figure 7: Plot of currents: $J^{k^+}$, $J^{k^-}$, $(J^{k^+} + J^{k^-})/2$, and $|J^{k^+} - J^{k^-}|$ with lines represent the theoretical results and symbols representing the Monte Carlo simulations.

## 4.4 Boundary layer analysis

A complete description of the density profile requires determining its behavior in the bulk as well as near the boundaries of the system. At the boundaries, the bulk solution typically fails to satisfy both boundary conditions, necessitating the introduction of an inner or boundary layer solution. We denote these inner solutions by $\rho^{k,\text{in}}$ and $\sigma^{k,\text{in}}$ for the $+$ and $-$ particle species, respectively, in lane $k$, where $k \in \{A, B\}$. This solution must satisfy two key conditions: (i) it must satisfy the boundary conditions that bulk solution fails to meet and (ii) it must asymptotically approach the bulk solution away from the boundary.

By enforcing these matching conditions, a combination of inner and outer solutions yield a complete description of the particle's density profile. The differential equations governing the bulk dynamics of positive and negative particles in a lane $k$, where $k \in \{A, B\}$, given by Eqs. (13) and (14) can be rewritten as:

$$\frac{d}{dx}\left[\frac{\epsilon}{2}\frac{d\rho^k}{dx} + \rho^k(\rho^k - 1)\right] = 0,$$
$$\frac{d}{dx}\left[\frac{\epsilon}{2}\frac{d\sigma^k}{dx} - \sigma^k(\sigma^k - 1)\right] = 0. \tag{46}$$

To derive the inner solution, the above governing equations are rewritten in terms of the rescaled spatial variable $\tilde{x} = \frac{x - x_0}{\epsilon}$, where $x_0$ denotes the position of the boundary layer and $\epsilon$ is a small parameter characterizing the boundary layer width. This rescaling is essential for capturing the rapid variation in particle density near the boundary, which the bulk equations cannot describe accurately. Under this transformation, the above equations are expressed in terms of $\tilde{x}$ as

$$\frac{d}{d\tilde{x}}\left[\frac{1}{2}\frac{d\rho^{k,in}}{d\tilde{x}} + \rho^{k,in}(\rho^{k,in} - 1)\right] = 0,$$
$$\frac{d}{d\tilde{x}}\left[\frac{1}{2}\frac{d\sigma^{k,in}}{d\tilde{x}} - \sigma^{k,in}(\sigma^{k,in} - 1)\right] = 0. \tag{47}$$

Upon integrating these equations with respect to $\tilde{x}$, we have

$$\frac{1}{2}\frac{d\rho^{k,in}}{d\tilde{x}} + \rho^{k,in}(\rho^{k,in} - 1) = C_1, \tag{48}$$

$$\frac{1}{2}\frac{d\sigma^{k,in}}{d\tilde{x}} - \sigma^{k,in}(\sigma^{k,in} - 1) = C_2, \tag{49}$$

where $C_1$ and $C_2$ are constants of integration, whose values are determined by utilizing the condition that the boundary layer solution must asymptotically approach to the outer solution as $\tilde{x}$ varies. As a result, $C_1$ and $C_2$ are given by

$$C_1 = \rho^k(\rho^k - 1), \quad C_2 = -\sigma^k(\sigma^k - 1), \tag{50}$$

where $\rho^k$ and $\sigma^k$ denote the outer or bulk solutions for positive and negative particles, respectively, in lane $k$. Substituting these values into Eqs. (48) - (49) and further solving the resulting first order differential equation, we obtain

$$\rho^{k,in}(\tilde{x}) = \frac{1}{2} + \frac{|2\rho^k - 1|}{2}\tanh\left(\frac{|2\rho^k - 1|\xi}{2}\right), \tag{51}$$

$$\sigma^{k,in}(\tilde{x}) = \frac{1}{2} - \frac{|2\sigma^k - 1|}{2}\tanh\left(\frac{|2\sigma^k - 1|\gamma}{2}\right), \tag{52}$$

where $\xi = 2\tilde{x} + k_1$ and $\gamma = 2\tilde{x} + k_2$. The constants $k_1$ and $k_2$ are uniquely determined by enforcing the boundary conditions on the inner solution. To determine the positions of boundary layers for each particle species in both lanes across all phases, we perform a fixed point analysis of the inner equations governing the behavior of boundary layer, as specified in Eqs. (48) and (49). By analyzing the stability of the fixed points and the structure of the corresponding stable and unstable manifolds, we identify how density trajectories connect boundary conditions to the bulk densities, thereby establishing the precise positions where boundary layers emerge in each phase.

635     We begin by identifying the fixed points in terms of the variables $(\rho^{k,\mathrm{in}}, \sigma^{k,\mathrm{in}})$ corre-
636 sponding to the spatial dynamical system governed by Eqs. (48) and (49). The fixed
637 points are determined by simultaneously solving $d\rho^{k,\mathrm{in}}/d\tilde{x} = 0$ and $d\sigma^{k,\mathrm{in}}/d\tilde{x} = 0$, yielding
638 four equilibrium points expressed as

$$
\begin{aligned}
(\rho_1^{k,in^*}, \sigma_1^{k,in^*}) &= (\rho^k, \sigma^k), \\
(\rho_2^{k,in^*}, \sigma_2^{k,in^*}) &= (\rho^k, 1 - \sigma^k), \\
(\rho_3^{k,in^*}, \sigma_3^{k,in^*}) &= (1 - \rho^k, \sigma^k), \\
(\rho_4^{k,in^*}, \sigma_4^{k,in^*}) &= (1 - \rho^k, 1 - \sigma^k).
\end{aligned}
\tag{53}
$$

639 To obtain the stability characteristics of each equilibrium points, we investigate the Jaco-
640 bian matrix $J$ corresponding to the dynamical system defined by Eqs. (48) and (49). Due
641 to the decoupled structure of these equations for a fixed lane $k$, the Jacobian simplifies
642 to a $2 \times 2$ diagonal matrix with diagonal entries given explicitly as $J_{11} = 2\rho^{k,\mathrm{in}} - 1$ and
643 $J_{22} = 1 - 2\sigma^{k,\mathrm{in}}$. The eigenvalues of this Jacobian govern the linear stability of the fixed
644 points. Specifically, a fixed point is classified as stable if all eigenvalues of $J$ are negative,
645 unstable if all are positive, and as saddle point when the eigenvalues have opposite signs.
646 The stability conditions for each of the fixed point, depending on the values of $\rho^k$, $\sigma^k$ are
647 summarized in the Table 1 and 2.

| Fixed point | $\max\{\rho^k, \sigma^k\} < 0.5$ | $\rho^k < 0.5 < \sigma^k$ | $\sigma^k < 0.5 < \rho^k$ | $\min\{\rho^k, \sigma^k\} > 0.5$ |
|---|---|---|---|---|
| $(\rho^k, \sigma^k)$ | Saddle | Unstable | Stable | Saddle |
| $(\rho^k, 1 - \sigma^k)$ | Unstable | Saddle | Saddle | Stable |
| $(1 - \rho^k, \sigma^k)$ | Stable | Saddle | Saddle | Unstable |
| $(1 - \rho^k, 1 - \sigma^k)$ | Saddle | Stable | Unstable | Saddle |

Table 1: Stability of the fixed points with respect to varying values of $\rho^k$ and $\sigma^k$.

| Fixed Point | $\rho^k = 0.5$ $\sigma^k < 0.5$ | $\rho^k = 0.5$ $\sigma^k > 0.5$ | $\rho^k < 0.5$ $\sigma^k = 0.5$ | $\rho^k > 0.5$ $\sigma^k = 0.5$ | $\rho^k = 0.5$ $\sigma^k = 0.5$ |
|---|---|---|---|---|---|
| $(\rho^k, \sigma^k)$ | $0, -$ | $0, +$ | $0, +$ | $0, -$ | $0, 0$ |
| $(\rho^k, 1 - \sigma^k)$ | $0, -$ | $0, +$ | $0, +$ | $0, -$ | $0, 0$ |
| $(1 - \rho^k, \sigma^k)$ | $0, +$ | $0, -$ | $0, -$ | $0, +$ | $0, 0$ |
| $(1 - \rho^k, 1 - \sigma^k)$ | $0, +$ | $0, -$ | $0, -$ | $0, +$ | $0, 0$ |

Table 2: Values and signs of the eigenvalues corresponding to fixed points when
either $\rho^k = 0.5$ or $\sigma^k = 0.5$.

648     The system may exhibit boundary layers near one or both boundaries $x = 0$ or $x = 1$,
649 which can be understood in the following ways:

650 • When a boundary layer forms at the left boundary of the system, i.e., $x = 0$, the corre-
651     sponding phase-space trajectories must connect the left boundary densities $(\rho^k(0), \sigma^k(0))$
652     to the bulk fixed point $(\rho^k, \sigma^k)$ as the rescaled coordinate $\tilde{x}$ increases. This necessitates
653     that the bulk solution functions as a stable fixed point, attracting trajectories originating
654     from the left boundary.

655 • If a boundary layer is localized at right end of the system, i.e., $x = 1$, the bulk solution
656     must instead act as an unstable fixed point, with trajectories diverging from it and
657     evolving toward the right boundary densities $(\rho^k(1), \sigma^k(1))$ as $\tilde{x}$ advances.

- In a scenario, when boundary layers form at both ends, the bulk solution $(\rho^k, \sigma^k)$ exhibits mixed stability characteristics: it acts as a stable fixed point for trajectories originating from $x = 0$ and as an unstable fixed point when approached for $x = 1$. Such a fixed point is identified as a saddle point, with a stable manifold that attracts trajectories from the left boundary and an unstable manifold that directs them toward the right boundary. Only when the fixed point possesses these contrasting stability properties, then the system sustains boundary layers at both ends while maintaining the common bulk solution.

We now examine the locations of boundary layers for each particle species in lane $k$, with $k \in \{A, B\}$, across all distinct existing phases of the system.

1. *Low density phases*: Consider a phase where the bulk densities of $+$ and $-$ particles in lane $k$ satisfy the condition $\rho^k < 0.5$ and $\sigma^k < 0.5$, corresponding to low-occupancy regimes such as the symmetric LD–LD/LD–LD or asymmetric L–L/L–L phases. For a boundary layer to exist at left boundary $x = 0$, trajectories in $(\rho - \sigma)$ phase space must approach to the bulk fixed point $(\rho^k, \sigma^k)$ as the rescaled coordinate $\tilde{x}$ increases. The associated streamline plot [see Fig. 8a] reveals that the only trajectory converging to this fixed point is along a vertical path with constant $\rho^k$ and varying $\sigma^k$. This trajectory represents the stable manifold of the fixed point, implying that the boundary condition $\rho^k(0) = \rho^k$ is naturally satisfied, while $\sigma^k(0) \neq \sigma^k$. As a result, no boundary layer forms for the $+$ species at $x = 0$.

In contrast, for a boundary layer to appear at the right boundary $x = 1$, there must exist phase space trajectories that repel from the bulk fixed point toward the boundary densities. The streamline plot in Fig.8a shows that the only such trajectory lies along the horizontal line, where $\sigma^k$ remains constant and $\rho^k$ varies indicating the absence of a boundary layer for the $-$ particle species at $x = 1$. In conclusion, in these phases, boundary layer forms at both ends, at $x = 1$ for $+$ particle species and at $x = 0$ for $-$ particle species in each lane as illustrated by the density profiles shown in Fig. 8b.

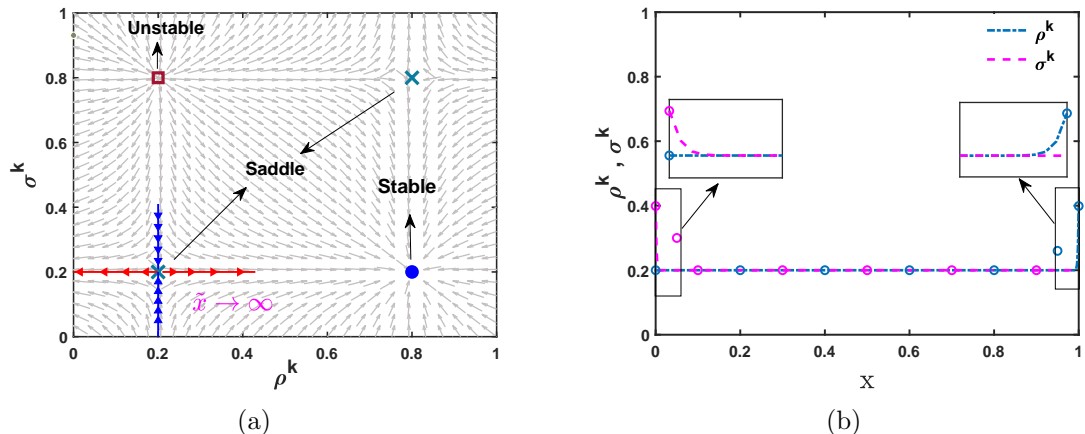

(a)                                      (b)

Figure 8: (a) Typical stream line plot and (b) corresponding density profile for low-low density phase (LD-LD/LD-LD or L-L/L-L). Insets in (b) display the enlarged view of the boundary layers observed. Symbols describe simulation results, while lines correspond to analytical findings.

2. *High-low density phases:* Let us now consider the phases in which one of the particle species in each lane exhibits a high density phase while the other remains in a low density phase, with $(\rho^k > 0.5$ and $\sigma^k < 0.5)$, or $(\rho^k < 0.5$ and $\sigma^k > 0.5)$. Considering

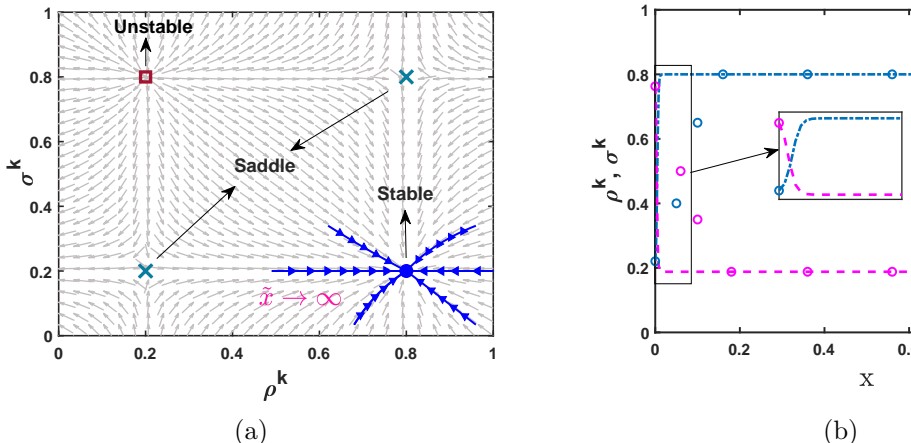

Figure 9: (a) Typical stream line plot and (b) corresponding density profile for high-low density phase (H-L/H-L). Insets in (b) display the enlarged view of the boundary layers observed. Symbols describe simulation results, while lines correspond to analytical findings.

the former case, the bulk fixed point $(\rho^k, \sigma^k)$ is globally stable, and all trajectories in the phase space asymptotically approach to this point as $\tilde{x} \to \infty$. This results in the formation of a boundary layer at the left boundary $(x = 0)$. The streamline plot in Fig. 9a, illustrates that all trajectories in the phase space converge toward the globally stable bulk fixed point. Notably, there exist no trajectory that diverges from this point as $\tilde{x} \to \infty$, ensuring that the right boundary condition is satisfied by the bulk densities. Consequently, boundary layers are formed only at $x = 0$ in both lanes for each species as affirmed in Fig. 9b.

3. *Shock–low density phase:* In this regime, the density profile of the $+$ particle species exhibits a discontinuous jump, forming a shock that connects two fixed points $(\rho^k, \sigma^k)$ to $(1 - \rho^k, \sigma^k)$ with $\rho^k < 1/2$ and satisfying both boundary conditions as a result of a discontinuous bulk profile. This implies no boundary layers are present for the $+$ species in either lane. In contrast, for $-$ particle species, which move from right to left, remain in the low density phase and satisfy the right boundary condition, i.e., $\sigma^k(1) = \sigma^k$. Consequently, no boundary layer forms at $x = 1$ for the $-$ species. However, a mismatch at the left boundary necessitates the formation of a boundary layer at $x = 0$.

4. *Maximal–low density phase*: In this phase, the $+$ particle species in both lanes exhibit maximal current behavior, with the bulk density fixed at $\rho^k = 0.5$. As evident from the streamline plot in Fig. 10a, There are some trajectories apart from the vertical one that converge to the bulk fixed point $(\rho^k, \sigma^k)$. This implies that at the left boundary $x = 0$, the conditions $\rho^k(0) \neq \rho^k$ and $\sigma^k(0) \neq \sigma^k$ hold. Consequently, the boundary condition at the left end is not satisfied for either particle species, thereby necessitating the formation of boundary layers at $x = 0$ for both particle species.

At the right boundary $x = 1$, as evident from Fig. 10a, the trajectory repels from the fixed point $(\rho^k, \sigma^k)$ corresponds to the horizontal direction, where $\sigma^k$ remains constant and $\rho^k$ varies. This implies that $\sigma^k(1) = \sigma^k$ while $\rho^k(1) \neq \rho^k$, indicating that the $-$ particles satisfy their boundary condition at $x = 1$ and thus do not form a boundary layer there. In conclusion, $+$ particles form boundary layers at both the boundaries whereas the $-$ particles form a boundary layer only at the left boundary $x = 0$ as

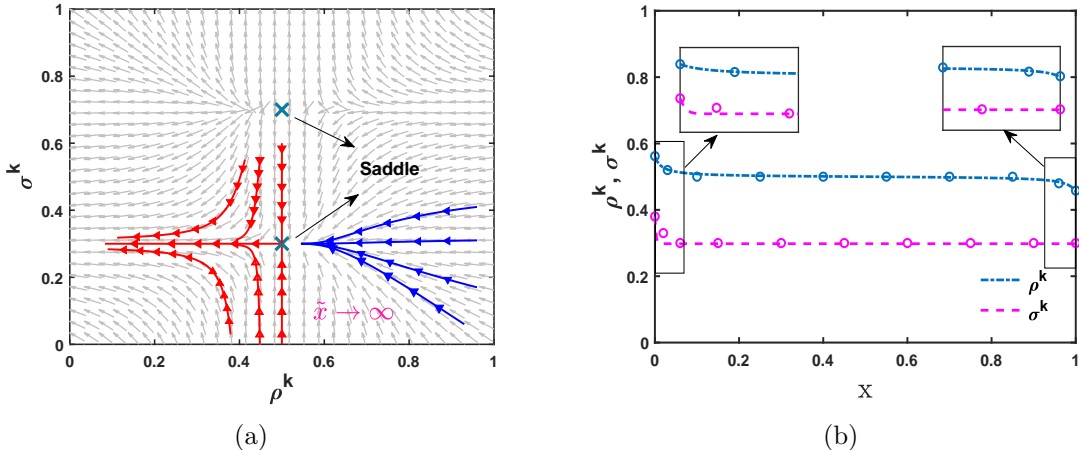

(a)                                                                   (b)

Figure 10: (a) Typical stream line plot and (b) corresponding density profile for maximal-low density phase (M-L/M-L). Insets in (b) display the enlarged view of the boundary layers observed. Symbols describe simulation results, while lines correspond to analytical findings.

displayed in Fig. 10b.

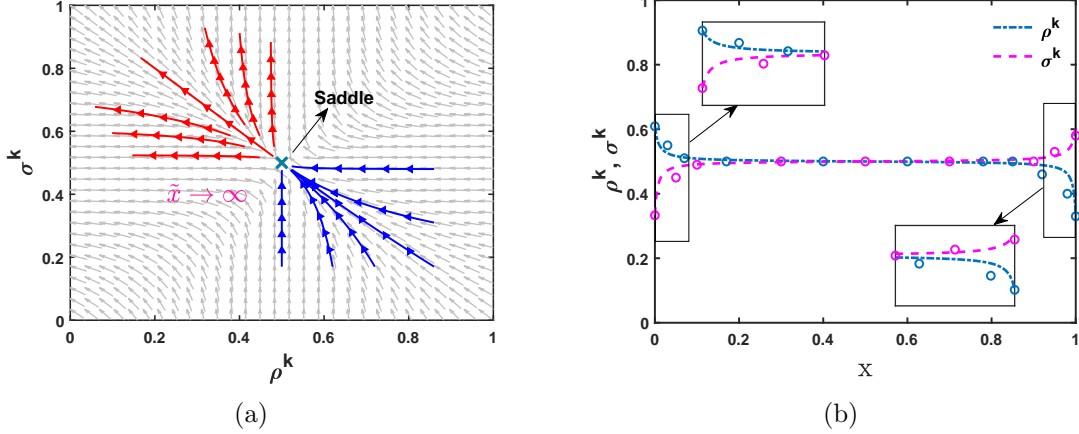

(a)                                                                   (b)

Figure 11: (a) Typical stream line plot and (b) corresponding density profile for maximal-maximal density phase (MC-MC/MC-MC or M-M/M-M). Insets in (b) display the enlarged view of the boundary layers observed. Symbols describe simulation results, while lines correspond to analytical findings.

5. *Maximal–maximal density phase:* In this phase, both the $+$ and $-$ particle species attain the maximal current phase, characterized by a bulk density of $\rho^k = \sigma^k = 0.5$. As evident from the streamline plot in Fig. 11a, trajectories originating in the region $\rho^k > 1/2$ and $\sigma^k < 1/2$ converge to the bulk fixed point $(\rho^k, \sigma^k)$ along the stable manifolds, while those from the region $\rho^k < 1/2$ and $\sigma^k > 1/2$ diverge from it. Due to this mixed stability, the fixed point fails to satisfy boundary conditions at both ends, i.e., $\rho^k(0) \neq \rho^k$, $\rho^k(1) \neq \rho^k$, $\sigma^k(0) \neq \sigma^k$, and $\sigma^k(1) \neq \sigma^k$. As a result, in this phase, boundary layers form at both boundaries for each particle species as shown in Fig. 11b.

## 4.5 Shock dynamics

A key consequence of imposing a constraint on the total number of positive and negative particles is the emergence of localized shocks in the system. In general, a shock phase is characterized by a discontinuous density profile, sharply separating two regions with distinct densities. In our system, shocks emerge in the density profiles of positive particles in both lanes, while negative particles remain in a low density state, giving rise to the asymmetric S–L/S–L phase. The characteristics of this localized shock, such as its position and height, are governed by parameters, including the intrinsic entry-exit rates and the filling factors. To scrutinize the behavior of the shock in the S-L/S-L phase under the condition $\mu^+ = \mu^-$, we fix the filling factor at $\mu^\pm = 0.94$ with $\alpha = 3$, while vary the exit rate $\beta$. The shock position, $x_w$, as determined by Eq. (36), along with the corresponding shock height, $\Delta$, are given by:

$$x_w = \frac{1}{1-2\beta}\left(1 - \beta - \mu^\pm + r^+\right), \quad \Delta = (1-\beta) - \alpha_{eff}^{k+}. \tag{54}$$

Here, $\alpha_{eff}^{k+}$ and $r^+$ are given by Eqs. (33) and (34), respectively. As $\beta$ increases and approaches the critical value of 0.31, the shock position $x_w$ shifts toward the right boundary at $x = 1$, leading to a transition from the S–L/S–L phase to the L–L/L–L phase [see Fig. 2c]. This shift is accompanied by a gradual reduction in the shock height $\Delta$ [see Fig. 13e], indicating a depletion of particles within the lanes.

When the entry parameter is fixed and $\beta$ is varied, the system exhibits a distinctive behavior in the phase diagram known as *back-and-forth transition* as observed for $\mu^\pm = 0.94$ [see Fig. 2c, 12a]. The term back-and-forth transition refers to a sequence of phase transitions observed in the stationary phase diagram, defined as follows: if the system undergoes a transition from phase P to phase Q and then subsequently returns to phase P, i.e. P → Q → P, while varying only a single parameter and keeping the remaining parameters fixed, this behavior is referred to as the back-and-forth transition [30,45,58]. In our system, this transition is observed in the phase diagram for $\mu^\pm = 0.94$, which reveals the emergence of an intermediate H-L/H-L phase embedded within the S-L/S-L phase region over a finite range of parameter values [see Fig. 2c]. To elucidate this behavior, we fix the parameters at $\alpha = 10$ and $\mu^\pm = 0.94$, and vary the exit rate $\beta$. As $\beta$ increases the system exhibit a phase transition from the S-L/S-L phase to the H-L/H-L phase and transition back to the S-L/S-L phase. A similar behavior is observed when the exit parameter is fixed at $\beta = 0.11$ and $\alpha$ is varied. The phase boundary between the S-L/S-L and H-L/H-L phases can be determined by solving the equation $x_w = 0$ which implies

$$\alpha = \frac{\beta^3 \mu^-(\beta-1)}{(\beta-1+\mu^-)\left(\beta^3 - 2\beta(\alpha_{eff}^{k-}-1)\alpha_{eff}^{k-} - (\alpha_{eff}^{k-}-1)^2(\alpha_{eff}^{k-})^2 + \beta^2\left((\alpha_{eff}^{k-})^2 - \alpha_{eff}^{k-}-1\right)\right)}, \tag{55}$$

where $\alpha_{eff}^{k-}$ is given by Eq. (33). It reveals that for $\mu^\pm = 0.94$, upon fixing either the entry or exit parameter and varying the other, the phase boundary exhibits a non-monotonic dependence on the varying parameters. The corresponding density profiles along with the shock position are shown in Figs. 13a and 13c, respectively. This unusual behavior of the system can be understood as follows: for fixed $\alpha = 10$ and $\mu^\pm = 0.94$, an increase in $\beta$ enhances the exit rates of both species, thereby increasing the respective reservoir densities. This in turn, elevates the effective entry rates $\alpha_{eff}^{k\pm}$ in both lanes ($k \in \{A, B\}$), leading to a significant accumulation of + particles. The resulting crowding hinders the exit of − particles, creating an imbalance that gives rise to shock formation in the density profiles of + species across both lanes. During this phase, the system initially maintains a balance between the entry and exit rates in both lanes, satisfying $\alpha_{eff}^{k+} = \beta$ for +

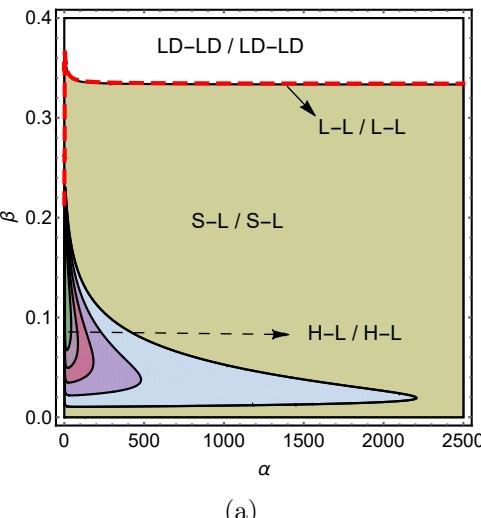
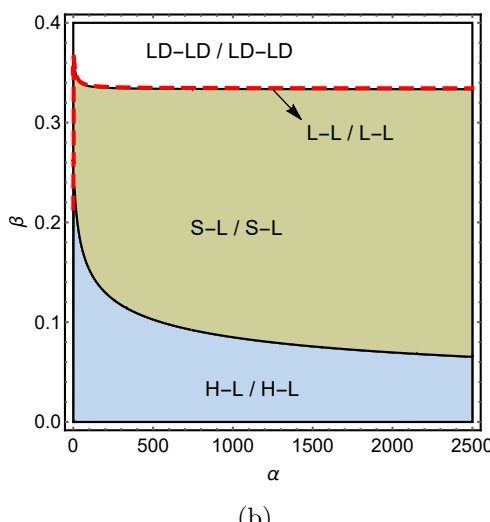

Figure 12: S-L/S-L and H-L/H-L phase region for different values of the filling factor $\mu^{\pm}$. (a) The innermost loop corresponds to $\mu^{\pm} = 0.94$, while the outermost boundary represents the case $\mu^{\pm} = 0.99$. The direction of arrow denotes the shift in H-L/H-L phase boundary as $\mu^{\pm}$ increases. (b) $\mu^{\pm} = 1$. Back-and-forth is observed for $\mu^{\pm} \in [0.94, 1)$.

species. However as $\beta$ increases, this balance is disrupted, and the shock associated with + particles shifts toward the left boundary. At the critical value $\beta = 0.0923$, the system undergoes a transition from the S–L/S–L phase to the H–L/H–L phase. In this phase, the effective entry rate of + particles exceeds the exit rate, leading to sustained accumulation. Upon further increase of $\beta$, this accumulation is progressively reduced, and the system transitions back to the S–L/S–L phase at $\beta = 0.1639$.

As illustrated in Fig. 2c, beyond the critical value of $\alpha = 21.3$, the back-and-forth transition disappears, and system remains in a shock phase for the + species. For all $\alpha > 21.3$, variation of exit rate $\beta$ induces a direct transition from the S-L/S-L phase to the L-L/L-L phase. As $\beta$ approaches the critical value of 0.347, the shock shifts toward the right boundary at $x = 1$, giving rise to an asymmetric low density phase. To delineate the parameter region supporting the back-and-forth transition, we examine its extent in the $\alpha-\beta$ parameter space for varying values of the symmetric filling factor $\mu^{\pm}$. This transition emerges at for $\mu^{\pm} > 0.933$ and persists over a finite range of $\alpha$. As $\mu^{\pm}$ increases, this range broadens, indicating that the transition occurs over an increasingly broader range of $\alpha$ [see Fig. 12a]. However at the critical value of $\mu^{\pm} = 1$, this transition disappears entirely from the system, as illustrated in Fig. 12b.

Let us turn our analysis to examine the behavior of shock in the system for $\mu^{+} \neq \mu^{-}$. To investigate this, the filling factors for + and − particles are fixed at $\mu^{+} = 0.96$ and $\mu^{-} = 0.8$, respectively. Phase diagram in Fig. 4b demonstrates that for asymmetric filling factors, the system exhibits a back-and-forth transition, characterized by a sequence of phase transitions: S–L/S–L $\rightarrow$ H–L/H–L $\rightarrow$ S–L/S–L, similar to the behavior observed in the symmetric case. To further illustrate this behavior under asymmetric filling, we fix $\alpha = 4$ with $\mu^{+} = 0.96$ and $\mu^{-} = 0.8$ while varying $\beta$. As shown in Fig. 4b, the shock associated with the + particle species shifts toward the left boundary and system undergoes a transition from the S–L/S–L phase to the H–L/H–L phase at the critical value $\beta = 0.065$. The corresponding density profiles, along with the shock position and height, are presented in Figs. 13b, 13d, and 13f, respectively. Upon further increasing $\beta$, the system reenters the S-L/S-L phase at $\beta = 0.2$, demonstrating a back-and-forth transition.

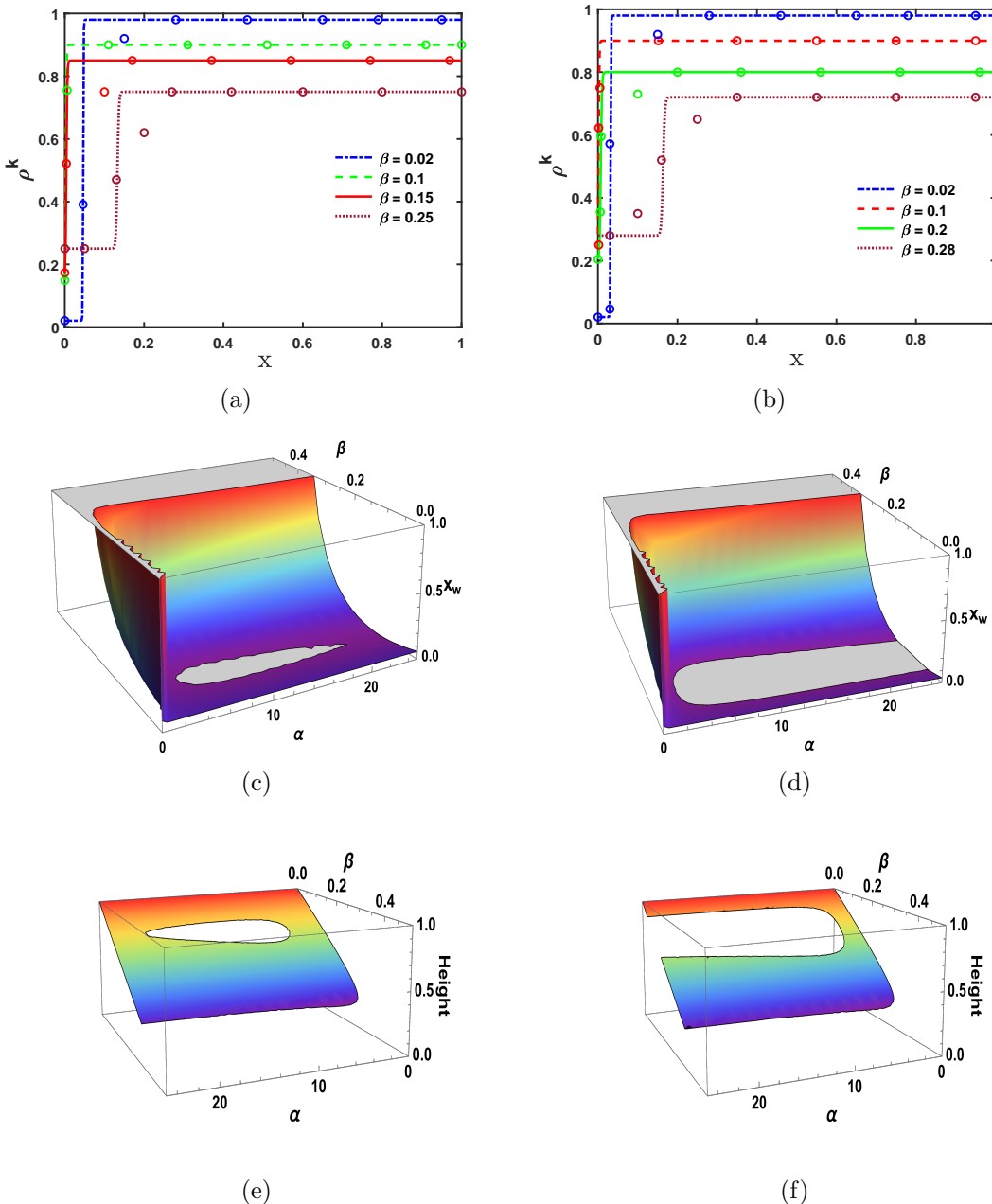

Figure 13: Subfigures (a), (c), and (e) correspond to the case $\mu^{\pm} = 0.94$, while (b), (d), and (f) correspond to $\mu^{+} = 0.96$ and $\mu^{-} = 0.8$. (a), (b) Density profiles for fixed $\alpha = 10$ and $\alpha = 4$, respectively, as $\beta$ varies. (c), (d) Shock position and (e), (f) shock height in the S-L/S-L phase, plotted as functions of $\alpha$ and $\beta$ for the respective cases.

As the filling factor values $\mu^{+}$ and $\mu^{-}$ approach infinity, the width of the S-L/S-L phase region decreases and eventually vanishes from the phase diagram indicating the absence of back-and-forth transitions in the limit of infinite particle reservoirs [see Fig. 4c].

## 4.6 Finite-size effects

In the theoretical analysis, the system is treated in the thermodynamic limit, $L \to \infty$, whereas simulations necessarily involve a finite system size. This discrepancy leads to ob-

servable finite-size effects, particularly near the system boundaries and within the shock regions, and may also influence the emergence of certain phases that are otherwise well-defined in the continuum limit. For TASEP models with bidirectional dynamics demonstrate that systems exhibiting symmetry breaking phenomena often display size-scaling dependencies [19, 26, 29, 32]. To examine the impact of finite lane size on the shock-low density phase in our model, we analyze the density profiles at fixed parameter values $(\alpha, \beta) = (1.5, 0.28)$ and $\mu^{\pm} = 0.94$, which displays S-L/S-L phase for different lane sizes $L$. As evident from Fig. 14a, as $L$ increases, the shock profiles in the S-L/S-L phase become progressively sharper, however the underlying phase remains qualitatively unchanged despite changes in lane size.

Now, consider the other asymmetric phase, L-L/L-L, which emerges in the system even at relatively low particle numbers for each species and remains persistent as the particle count increases. With in the mean-field framework, this phase is confined to a curve while simulations reveal its presence over a significantly extended region. As reported in earlier studies [34, 59], the finite length of the lanes significantly influences the properties of the L-L/L-L phase. It has been observed that as the system size increases, the width of the asymmetric low density phase region gradually diminishes, indicating that this region may either reduce to a line or vanish entirely in the thermodynamic limit. To investigate this effect, we have plotted the region width $\delta$ with respect to $\beta$ for fixed parameter values $\alpha = 1.5$ and $\mu^{\pm} = 0.6$. As depicted in Fig. 14b, the region width $\delta$ decreases with increasing system size $L$, eventually shrinking to a narrow region around $L \approx 3000$, validating our theoretical outcomes.

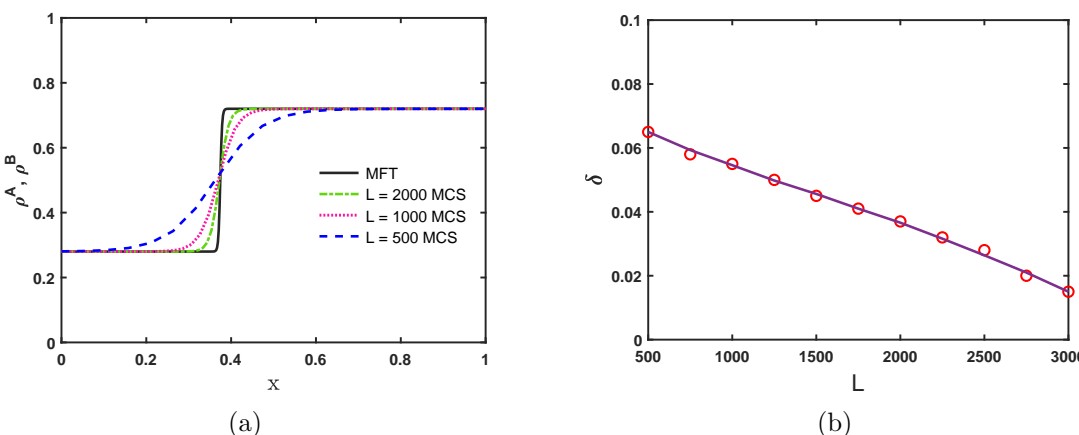

Figure 14: (a) Effect of finite system size on the shock-low density (S-L/S-L) profile for fixed parameters $(\alpha, \beta, \mu^{\pm}) = (1.5, 0.28, 0.94)$ and varying values of lane length $L$. (b) Region width ($\delta$) of low-low density asymmetric (L-L/L-L) phase with respect to $\beta$ for fixed $\alpha = 1.5$ and $\mu^{\pm} = 0.6$. The solid curve represents a best-fit polynomial to the discrete simulation data (indicated by red markers) up to $L \approx 3000$.

# 5 Summary and conclusion

In this work, we analyze a theoretical model that captures the bidirectional transport of finite particles in a two-lane system with constraint entrances, inspired by systems such as cargo vesicle transport by motor proteins along parallel microtubules and vehicular flow on narrow roadways. This model is formulated as a two-species, multi-lane bidirectional

TASEP, incorporating distinct particle reservoirs for each particle species. The entrance of each particle species into the lanes is regulated by the occupancy of its respective reservoir. The total particle number of each species is conserved throughout the system and is quantified by a corresponding filling factor. By employing a mean-field theoretical framework, we systematically examine how the system's intrinsic dynamics governs its steady-state behavior, with particular emphasis on key features such as phase diagrams, density profiles, phase boundaries, and phase transitions. These analytical findings are further validated through extensive stochastic simulations performed using the Gillespie algorithm and finite difference scheme.

To investigate the impact of coupling the lanes to two distinct reservoirs, we consider two cases: (i) a symmetric case, where the filling factors for both species are identical, and (ii) an asymmetric case, where they differ. In the symmetric case, the system exhibits up to five distinct stationary phases, out of which, two are symmetric: LD–LD/LD–LD and MC–MC/MC–MC, while the remaining three are asymmetric: L–L/L–L, H–L/H–L, and S–L/S–L. Despite the identical dynamics of the two particle species, spontaneous symmetry breaking is observed, persisting even at very small value of the filling factor. To further investigate this phenomenon, we analyze particle density histograms generated from Monte Carlo simulations. A key consequence of coupling the system to finite reservoirs is the emergence of localized shocks. In particular, the asymmetric S–L/S–L phase that emerges exclusively under finite resource conditions and is absent in systems with infinite reservoirs. The number of distinct phases in the phase diagram exhibits a non-monotonic dependence on the common filling factor: it initially increases from 3 to 5 and then decreasing back to 4 as the filling factor becomes larger.

In case of unequal filling factors, the system exhibits upto five distinct stationary phases. Due to the inherent asymmetry between the two reservoirs, symmetric phases cannot persist. The introduction of asymmetric filling factors significantly alters the phase structure, affecting it both quantitatively and qualitatively. A notable feature is the emergence of maximal-low density (M-L/M-L) and maximal-maximal (M-M/M-M) phases which do not appear in analogous bidirectional models with unlimited resources [57]. Even in this case, the number of distinct phases in the phase diagram displays a non-monotonic variation with varying filling factor values. One of the most remarking feature is the emergence of back-and-forth transitions in the phase diagram, which occur for both symmetric and asymmetric filling factor conditions, a characteristics not previously reported in two-species interacting system with narrow entrances [57].

To investigate the characteristics of shock phase in more detail, we analyze the shock position, its height, along with the particle density profiles as functions of the relevant system parameters. Explicit calculations of phase boundaries and density profiles are presented for all feasible phases, accompanied by clear physical interpretations that elucidate the underlying mechanisms driving the theoretical results. Furthermore, a detailed boundary layer analysis is performed to derive the inner solutions near the system boundaries. In addition, a fixed point analysis of the associated dynamical system is conducted to accurately predict the positions of the boundary layers, based on the stability properties of the corresponding fixed points.

This model aims to capture key features of transport processes encountered in both biological and physical contexts. In intracellular transport, motor proteins like kinesin-1 and dynein-1 move in opposite directions along parallel microtubule filaments, competing for access to spatially constrained and mutually exclusive binding sites. The inter-lane entrance constraint incorporated in our model effectively mimics such inter-filament interactions, where steric hindrance and lane-dependent accessibility play crucial roles. Additionally, the coupling to finite particle reservoirs reflects the limited availability of motor

proteins and cargos in confined cellular environments, providing a realistic representation of resource constrained biological transport. Similarly, in physical systems such as bidirectional vehicular or pedestrian traffic on narrow multi-lane roads or pathways, a vehicle's ability to enter or switch lanes can depend on the state of neighboring lanes, while a finite supply of incoming traffic reflects realistic boundary constraints. The present model thus offers a minimal yet versatile framework for exploring how local microscopic exclusion rules, combined with global resource constraints govern macroscopic transport phenomena in driven nonequilibrium systems. The findings of this study provide useful insights for the regulation and control of transport processes in complex bidirectional systems featuring bottlenecks. Moreover, they may contribute in the development of effective strategies for managing congestion and enhancing flow efficiency in a wide range of biological and engineered transport networks.

It is worth emphasizing that the analysis of the model in the case of $q \neq 1$ becomes considerably more intricate, and the insights obtained in the $q = 1$ case are not directly applicable [29]. Addressing this limitation remains a subject for our future investigation. Furthermore, the model can be extended to include additional dynamical processes, such as particle attachment and detachment on the lanes and lane coupling, which may introduce new steady-state features and further enrich the nonequilibrium phase behavior of the system.

# A    Gillespie algorithm

Let the state of the system at time $t$ be described by:

$$\boldsymbol{\gamma}^A = \left[\gamma_1^A, \gamma_2^A, \ldots, \gamma_L^A\right], \quad \text{(Lane A)},$$

$$\boldsymbol{\gamma}^B = \left[\gamma_1^B, \gamma_2^B, \ldots, \gamma_L^B\right], \quad \text{(Lane B)}.$$

Here, $\gamma_i^k \in \{1, 0, -1\}$, where 1, 0 and $-1$ correspond to site occupied by a $+$ particle, $-$ particle and empty site, respectively. The possible elementary events on each lane are

1. Entry and exit of $+$ or $-$ particles at boundaries.

2. Forward hopping of $+$ or $-$ particles, subject to exclusion.

3. Exchange events between $+$ and $-$ particles on adjacent sites.

In our system, one can identify $N = 14$ such possible events. For each event $j$ ($j \leq N/2$ correspond to lane A and remaining for lane B), we associate two arrays:

- $a_j$: stores the rate for each possible event $j$.

- $b_j$: stores the site index (or indices) associated with event $j$.

For any initial lane configuration $(\gamma^A, \gamma^B)$, the next event is chosen stochastically among the $N$ possible events according to their rates $a_j$. Once an event is selected, the state vectors $\gamma^A$ or $\gamma^B$ is updated accordingly, following the Gillespie algorithm outlined below.

1. **Input**: initial time $t = 0$; state $\gamma^k \in \{0, 1, -1\}^L$, $k \in \{A, B\}$; rates $\alpha, \beta$ and final time $t_{max}$.

2. **while** $t < t_{max}$, **do**

3. For each event $j$, compute $a_j$ and $b_j$.

4. **Define** $a_0 := \sum_{j=1}^{N} \sum_{s=1}^{\text{length}(a_j)} a_j(s)$.

5. Choose $r_1$, $r_2 \sim \text{Uniform}(0,1)$.

6. **Determine next event:**

$$b_n(m_n) = \min\left\{ n + m_j \middle|\ r_1 a_0 \leq \sum_{j=1}^{n} \sum_{s=1}^{m_j} a_j(s) \right\}$$

where $n \leq N$ and $m_n \leq \text{length}(a_j)$. If $n \leq N/2$, event $n$ occurs at site $b_n(m_n)$ in lane A, otherwise lane B.

7. **Determine next event time:** $dt = \frac{1}{a_0} \ln\left(\frac{1}{r_2}\right)$.

8. Update time $t \to t + dt$ and state $(\gamma^A, \gamma^B)$ according to the selected event.

9. end **while**

## B   Numerical scheme

Here we provide an alternative method for obtaining the density profiles using numerical techniques to complement the analytical treatment presented in Sec. 3. This approach, although less elaborate in formulation, yields exact numerical solutions and offers practical advantages in terms of implementation. A key advantage of this method lies in its adaptability: modifications to the microscopic dynamics can be seamlessly incorporated by updating the master equation, making it well-suited for exploring generalized variants of the model. In particular, when the analytical analysis becomes intractable, the numerical scheme remains fully applicable and continues to provide reliable insights into the system's stationary behavior.

The governing equations for the system given in Eqs. (11) and (12) are discretized using a finite difference scheme, where spatial and temporal derivatives are approximated by central and forward differences, respectively. The spatial domain is discretized with grid spacing $\Delta x = 1/L$, where $L$ is finite and the time step $\Delta t$ is chosen to satisfy the stability condition $\epsilon \Delta t \leq 2\Delta x^2$. We denote the densities of $+$ and $-$ particle species in lane $k \in \{A, B\}$ at spatial index $i$ and time step $j$ by $\rho_{i,j}^{k}$ and $\sigma_{i,j}^{k}$, respectively. The resulting discretized equations governing the bulk dynamics of both particle species are given by:

$$\begin{bmatrix} \rho_{i,j+1}^{k} \\ \sigma_{i,j+1}^{k} \end{bmatrix} = \begin{bmatrix} \rho_{i,j}^{k} \\ \sigma_{i,j}^{k} \end{bmatrix} + \frac{\epsilon \Delta t}{2\Delta x^2} \begin{bmatrix} \rho_{i,j+1}^{k} - 2\rho_{i,j}^{k} + \rho_{i-1,j}^{k} \\ \sigma_{i,j+1}^{k} - 2\sigma_{i,j}^{k} + \sigma_{i-1,j}^{k} \end{bmatrix} + \frac{\Delta t}{2\Delta x} \begin{bmatrix} (2\rho_{i,j}^{k} - 1)(\rho_{i+1,j}^{k} - \rho_{i-1,j}^{k}) \\ (1 - 2\sigma_{i,j}^{k})(\sigma_{i+1,j}^{k} - \sigma_{i-1,j}^{k}) \end{bmatrix}. \quad \text{(B.1)}$$

Owing to the specific dynamics of the model, inter-lane interactions occur exclusively at the boundaries, i.e., at sites $i = 1$ and $i = L$. Consequently, the bulk equations described above are not valid at these boundary sites. Instead, the dynamics at the boundaries must be treated separately using the boundary current expressions given in Eq. (16), leading to the following set of boundary equations:

$$\begin{bmatrix} \rho_{1,j+1}^{A} \\ \rho_{1,j+1}^{B} \end{bmatrix} = \begin{bmatrix} \rho_{1,j}^{A} \\ \rho_{1,j}^{B} \end{bmatrix} + \Delta t \alpha \left( 1 - \frac{\sum_{i=1}^{L}(\rho_{i,j}^{A} + \rho_{i,j}^{B})}{N^{\text{tot}^+}} \right) \begin{bmatrix} (1 - \sigma_{1,j}^{B})(1 - \rho_{1,j}^{A} - \sigma_{1,j}^{A}) \\ (1 - \sigma_{1,j}^{A})(1 - \rho_{1,j}^{B} - \sigma_{1,j}^{B}) \end{bmatrix}$$

$$- \Delta t \begin{bmatrix} \rho_{1,j}^A (1 - \rho_{2,j}^A) \\ \rho_{1,j}^B (1 - \rho_{2,j}^B) \end{bmatrix}, \tag{B.2}$$

$$\begin{bmatrix} \rho_{L,j+1}^A \\ \rho_{L,j+1}^B \end{bmatrix} = \begin{bmatrix} \rho_{L,j}^A \\ \rho_{L,j}^B \end{bmatrix} + \Delta t \begin{bmatrix} \rho_{L-1,j}^A (1 - \rho_{L,j}^A) \\ \rho_{L-1,j}^B (1 - \rho_{L,j}^B) \end{bmatrix} - \Delta t \beta \begin{bmatrix} \rho_{L,j}^A \\ \rho_{L,j}^B \end{bmatrix}. \tag{B.3}$$

Similar equations can be written for negative particle as well.

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
