# Peer review of "Coupled dynamics of resource competition and constrained entrances in a multi-lane bidirectional exclusion process"

_SciPost Physics Core_

## Round 1 · Referee Report · Anatoly Kolomeisky (Referee 1) · 2025-10-7

Strengths

1) Important scientific subject. 2) Multi-disciplinary application. 3) Comprehensive analysis that combines analytical, numerical calculations, and boundary layer analysis. 4) Excellent explanations of details and calculations. 5) Well-written manuscript that can be easily followed.

Weaknesses

None

Report

This is an outstanding theoretical work that combines analytical calculations and computer simulations to investigate a complex bidirectional transport system stimulated by biological observations. The paper is well-written, and the results are clearly explained. The authors did an excellent job of explaining the physical meaning of the results. It is definitely an important advancement in the field.

Requested changes

I have several minor suggestions for improvement: 1) Eq. (3) - It is the simplest assumption about the entrance rates, but it is not the only one. Will the results change if other choices are made? Some brief discussions might be useful here. 2) I would add more discussion on the expectation of what might happen when q<1. Specifically, if mean-field theory could capture the processes well. 3) I would mention that the success of mean-field theory for this model is because the coupling (correlations) are very local (only at the entrances).

Recommendation

Publish (surpasses expectations and criteria for this Journal; among top 10%)

  • validity: top
  • significance: top
  • originality: high
  • clarity: top
  • formatting: perfect
  • grammar: perfect

Author:  Arvind Kumar Gupta  on 2025-10-14  [id 5923]

(in reply to Report 1 by Anatoly Kolomeisky on 2025-10-07)

Please see attachment.

Attachment:

Scipost_Rebuttal__reviewer_1.pdf

---

## Round 1 · Referee Report · Anonymous (Referee 2) · 2025-10-8

Strengths

  1. The paper is a complete study of the phases of a transport model as a function of its parameters.
  2. The study includes both mean field theory and simulations.
  3. The paper is well written and mostly complete.
  4. This is one more example of a phase diagram of a transport model that can be added to a catalogue of phase diagrams of transport models.

Weaknesses

  1. There is very little new from the methodological side.
  2. There does no appear to be anything qualitatively new with respect to other transport models studied in the literature.

Report

I recommend to publish the paper after the revisions below have been considered. Although the work is not orginal, the research is well done and reported. Given that the results exist, I see no reason not to publish them.

Requested changes

  1. Please explain in more detail how the right-hand sides in (18) are obtained (i.e. the second equality signs)

  2. Section 3.2: It is said "numerical investigations corroborated by Monte Carlo Simulations confirm the absence of several theoretical feasible phases".

Could you please elaborate on this? Does this imply that there parameter regimes for which multiple phases can exist, but only one is observed?

  1. At present figures captions say "Solid lines represent theoretical predictions". It would be helpful if this can be made more precise, and the captions refer to specific equations in the main text that have been used to plot the solid lines.

  2. Figure 13 and 14: Figure 13 shows that for beta=0.25 and alpha=10 or beta=0.28 and alpha=4 the theory deviates significantly from the numerical simulations. Figure 14 seems to indicate that this is a finite size effect. However, Figure 14 is not showing resuls for th same parameters as Figure 13. Therefore, this is inconclusive. I think the authors should show a figure 14 for exactly the same parameters as figure 13, and overlay it with theoretical results, demonstrating that numerics converges towards theory. Otherwise, it remains inconclusive that the shock is localised as claimed in the paper.

  3. In the legend of figure 14: it is not clear what is meant by MCS.

Recommendation

Ask for major revision

  • validity: high
  • significance: low
  • originality: low
  • clarity: high
  • formatting: perfect
  • grammar: perfect

Author:  Arvind Kumar Gupta  on 2025-10-14  [id 5922]

(in reply to Report 2 on 2025-10-08)

Please see attachment.

Attachment:

Scipost_Rebuttal__reviewer_2.pdf

---

## Editorial Decision

accepted_in_target_journal